# GIFT: Gradient-aware Immunization of diffusion models against malicious Fine-Tuning with safe concepts retention

## Abstract

We present GIFT: a Gradient-aware Immunization technique to defend diffusion models against malicious Fine-Tuning while preserving their ability to generate safe content. Existing safety methods, such as safety checkers, are easily bypassed, and concept erasure methods fail under adversarial fine-tuning. GIFT addresses this by framing immunization as a bi-level optimization problem: the upper-level objective degrades the model's ability to represent malicious concepts using representation noising and maximization, while the lower-level objective preserves performance on safe data. GIFT achieves robust resistance to malicious fine-tuning while maintaining safe utility. Experimental results show that GIFT significantly impairs the model's ability to re-learn malicious concepts while maintaining performance on safe content, offering a promising direction for creating inherently safer generative models resistant to adversarial fine-tuning attacks.
Warning: This paper contains NSFW content. Reader discretion is advised.

## 1 Introduction

Text-to-image (T2I) models have emerged as powerful generative tools capable of producing high-quality images faithful to input prompts (Rombach et al., 2022; Podell et al., 2023; Imagen-Team-Google et al., 2024; Ramesh et al., 2021). However, their accessibility and adaptability make them vulnerable to malicious fine-tuning, where adversaries adapt pre-trained models to generate harmful or copyrighted content. Methods like DreamBooth (Ruiz et al., 2023), LoRA (Hu et al., 2022) , and Textual Inversion (Gal et al., 2023) enable this adaptation with minimal resources and without needing to train from scratch. This vulnerability persists even when existing safety mechanisms, such as safety checkers (Rando et al., 2022) or concept erasure methods (Gandikota et al., 2023; 2024; Gong et al., 2024), are in place, as they can be bypassed (Yang et al., 2024; Zhang et al., 2024b; Gao et al., 2024), disabled, or undone through lightweight adaptation techniques. This creates a significant risk: once a model is open-sourced, it becomes difficult to guarantee its continued alignment with safety goals. Current defenses either degrade the model's generative capabilities or fail to withstand adversarial fine-tuning.

While safety checkers and licensing agreements offer a first line of defense (CompVis, 2022), they are not an inherent property of T2I models and are easily circumvented (Rando et al., 2022). To enhance the inherent safety of T2I models, concept erasure techniques have been proposed to remove undesirable concepts by modifying the model's internal representations. Although such techniques can suppress the generation of undesired concepts, they are vulnerable to circumvention (Pham et al., 2024; Zhang et al., 2024b). Moreover, as we show in our experiments, simple fine-tuning can reintroduce the erased concepts, undermining the long-term effectiveness of concept erasure methods as a safety mechanism.

To address the limitations of concept erasure and defend against its circumvention, model immunization has been proposed as a proactive defense against malicious fine-tuning of T2I models. IMMA (Zheng & Yeh, 2024), for example, introduces a bi-level optimization approach inspired by MAML (Finn et al., 2017), aiming to learn poor model initializations that hinder adaptation to undesirable concepts. By simulating the fine-tuning process, IMMA makes it more difficult for adversaries to reintroduce malicious content through fine-tuning. However, IMMA's framework sig-

nificantly compromises the model's performance on safe concepts, degrading both its generative quality and its ability to be fine-tuned for safe applications as we will show in our experiments.

To this end, we propose **GIFT**—a **G**radient-aware **I**mmunization framework to defend T2I diffusion models against malicious **F**ine-**T**uning while preserving their ability to generate safe content. Inspired by IMMA (Zheng & Yeh, 2024) and MAML (Finn et al., 2017), we formulate GIFT as a bi-level optimization problem: the lower-level task minimizes a *prior preservation* objective to retain performance on safe concepts, while the upper-level task minimizes an *immunization* objective that prevents adaptation to malicious concepts. The *immunization* objective is composed of two parts: (1) a *loss maximization term* and (2) a *representation noising term* Rosati et al. (2024).

We demonstrate that immunizing a model with GIFT significantly impairs its ability to re-learn malicious content while maintaining generative ability across a wide range of safe concepts. Our evaluation covers several concept categories which are treated in turn as malicious—including objects, art styles, and NSFW content—and considers multiple fine-tuning strategies, *e.g.*, LoRA and DreamBooth. Our main contributions are:

- We propose GIFT, a novel framework that immunizes text-to-image diffusion models against malicious fine-tuning while preserving their generative utility on safe concepts.

- We formulate immunization as a bi-level optimization problem where the lower-level task uses a prior preservation loss to maintain generation quality on safe concepts, and the upper-level task employs an immunization loss to resist adaptation to malicious ones.

- We conduct extensive experiments across diverse concept types (objects, art styles, and NSFW content), demonstrating that GIFT outperforms existing baselines (ESD, IMMA) in resisting malicious fine-tuning while preserving safe model utility.

## 2  RELATED WORK

The advancements of text-to-image (T2I) generative models, such as Stable Diffusion (Rombach et al., 2022), have democratized content creation but have also introduced significant risks associated with their misuse. A key concern is the vulnerability of these models to malicious fine-tuning, where bad actors can adapt pre-trained models to generate harmful, copyrighted, or otherwise undesirable content, prompting growing interest in developing safeguards for T2I models.

Existing methods to mitigate these risks can be broadly categorized. One line of work focuses on **concept erasure**, aiming to remove specific concepts from a pre-trained model. Erased Stable Diffusion (ESD) (Gandikota et al., 2023) fine-tunes model weights using textual descriptions of the undesired concept to prevent the model from generating it. Other methods explore unlearning by modifying specific model components like the text encoder or attention layers (Kumari et al., 2023a; Zhang et al., 2024a), sometimes using few-shot unlearning techniques (Wu et al., 2025), by adding lightweight eraser modules (Gong et al., 2024), or through data unlearning (Alberti et al., 2025).

While effective at removal, some erasure techniques can be circumvented by further fine-tuning (Pham et al., 2024), as the underlying knowledge might not be entirely eliminated or can be easily relearned (Zhou et al., 2024; Zhang et al., 2024b). Additionally, a significant challenge is preserving the model's utility on unrelated concepts, as aggressive erasure can lead to "catastrophic forgetting" of desired knowledge (Tian et al., 2024; Xu et al., 2023). Some recent works attempt to address this by focusing on concept-localized regularization or mitigating conflicting gradients during unlearning (Patel & Qiu, 2025; Wu et al., 2025).

Another paradigm is **model immunization**, which seeks to make the model inherently resistant to adaptation towards malicious concepts before it is released. IMMA (Immunizing text-to-image Models against Malicious Adaptation) (Zheng & Yeh, 2024) proposes learning model parameters that are difficult for adaptation methods to fine-tune on malicious content, framed as a bi-level optimization problem. While IMMA demonstrates effectiveness against various adaptation methods like LoRA (Hu et al., 2022), Textual Inversion (Gal et al., 2023), and DreamBooth (Ruiz et al., 2023), it can be overly aggressive, potentially degrading the model's performance on safe, unrelated concepts. Other defense strategies include methods akin to data poisoning (*e.g.*, Glaze (Shan et al., 2023), which protects artistic styles from mimicry).

Other defense approaches focus on **safe decoding or generation**, often by modifying the diffusion process (Schramowski et al., 2023) or employing external classifiers and adaptive guards to filter outputs (Yoon et al., 2025; Wang et al., 2024). However, these can sometimes be bypassed by users with white-box access to the model or through carefully crafted adversarial prompts and jailbreaking methods (Rando et al., 2022; Yang et al., 2024; Gao et al., 2024; Liu et al., 2024b).

Techniques from the Large Language Model (LLM) domain are also being explored and adapted. Representation Noising (RepNoise) (Rosati et al., 2024), for instance, has been proposed as a defense mechanism against malicious fine-tuning in LLMs by removing information about representations of malicious concepts across model layers, making them difficult to recover. GIFT draws inspiration from this by adapting representation noising to T2I models.

Unlike some erasure methods that can be easily circumvented (Pham et al., 2024; Zhang et al., 2024b), and in contrast to immunization methods like IMMA (Zheng & Yeh, 2024) that may severely degrade general utility, GIFT aims for a better trade-off. GIFT's bi-level formulation helps to prevent the immunization objective from detrimentally affecting the prior preservation objective.

## 3 METHODOLOGY

### 3.1 PROBLEM FORMULATION

Our goal is to prevent adaptation methods from reintroducing malicious concepts into a pre-trained T2I diffusion model. We formulate this as a bi-level optimization problem with two objectives: (1) immunization against malicious concepts and (2) preservation of model performance on safe ones. We define "malicious" and "safe" concepts in a highly context-dependent way. A "malicious" concept is simply any concept that we wish to prevent the model from generating, while a "safe" concept refers to any other concept in that scenario. As an example, while images of some particular stuffed animal are ordinarily safe, the company producing that product may not want people to generate images of their product for reasons of copyright or repute.

Let $\theta$ represent the U-Net parameters of a pre-trained T2I model (*e.g.*, Stable Diffusion (Rombach et al., 2022)), and $\psi \subset \theta$ denote the subset of parameters corresponding to cross-attention layers. Let $(x_m, c_m) \in D_M$ and $(x_s, c_s) \in D_S$ denote image–text pairs from the malicious and safe datasets, respectively. We aim to derive an immunized model $\theta^I$ that resists adaptation to malicious concepts under any subsequent fine-tuning while maintaining its ability to learn/generate safe concepts.

### 3.2 BI-LEVEL OPTIMIZATION FRAMEWORK

The authors of IMMA (Zheng & Yeh, 2024) employ a meta-learning algorithm inspired by MAML (Finn et al., 2017) to immunize T2I models. They simulate malicious adaptation steps by minimizing the adaptation loss in the lower-level task, while maximizing the same loss in the upper-level task to achieve immunization. We employ a bi-level optimization framework to immunize a T2I model against malicious concepts while retaining performance on safe data. We define the upper-level task as the immunization objective over $D_M$ and the lower-level task as the prior preservation objective over $D_S$. We show our algorithm at 1.

To perform optimization, we compute parameters $\theta^*$ via a gradient step using the lower-level task on $D_S$, followed by an optimization step of said parameters using the upper-level task on $D_M$. We formulate this as the following bi-level optimization problem:

$$\underbrace{\psi^I = \arg\min_{\psi \subset \theta^*} \mathcal{L}_{\text{immunize}}(x_m, c_m; \theta^*)}_{\text{upper-level task}} \quad \text{where} \quad \underbrace{\theta^* = \arg\min_{\theta} \mathcal{L}_{\text{prior}}(x_s, c_s; \theta)}_{\text{lower-level task}}. \tag{1}$$

In the upper-level task, we minimize the *immunization loss* $\mathcal{L}_{\text{immunize}}$ with respect to $\psi$ (*i.e.*, cross-attention layers). This encourages the model to resist adapting to malicious concepts from the malicious dataset $D_M$. In the lower-level task, we minimize the *prior preservation loss* $\mathcal{L}_{\text{prior}}$ with respect to the U-Net parameters $\theta$, which includes $\psi$. This selection reflects the intuition that cross-attention layers play a central role in encoding and manipulating concepts (Liu et al., 2024a), while optimizing the U-Net in the inner loop ensures that the model adapts safely while incorporating the immunization updates.

To further explore how this bi-level setup benefits the immunization approach more than naive addition of all losses, we examine intermediate gradient updates. We implement the bi-level scheme in Eq. equation 1 by iterating the gradient updates:

$$\theta' = \theta - \alpha_P \nabla \mathcal{L}_P(\theta) \quad \text{and} \quad \psi'' = \psi' - \alpha_I \nabla \mathcal{L}_I(\psi'), \tag{2}$$

where $\psi' \subset \theta'$, and we abbreviate $\mathcal{L}_{\text{prior}}(x_s, c_s; \theta) = \mathcal{L}_P(\theta)$ and $\mathcal{L}_{\text{immunize}}(x_m, c_m; \theta) = \mathcal{L}_I(\psi)$. We then use the Taylor Series expansion for $\nabla \mathcal{L}_I(\psi')$ as seen in Eq equation 3, which gives us the total update for $\psi''$ in Eq equation 4. A full derivation can be found in Appendix A.

$$\nabla \mathcal{L}_I(\psi') \approx \nabla \mathcal{L}_I(\psi) - \alpha_P \nabla^2 \mathcal{L}_I(\psi) \nabla_\psi \mathcal{L}_P(\theta) \tag{3}$$

$$\psi'' \approx \psi - \alpha_P \nabla_\psi \mathcal{L}_P(\theta) - \alpha_I \nabla \mathcal{L}_I(\psi) + \alpha_P \alpha_I \nabla^2 \mathcal{L}_I(\psi) \nabla_\psi \mathcal{L}_P(\theta) \tag{4}$$

The final equation shows that our current immunization gradient update is equivalent to doing an update in the *prior preservation* direction plus an update in the *immunization* direction plus an *additional* term. That term is the directional curvature of $\mathcal{L}_I$ along $\nabla_\psi \mathcal{L}_P(\theta)$, which is crucial for our approach. This term adds a second order correction which helps coordinate the gradient descent so that minimizing $\mathcal{L}_I$ does not make minimizing $\mathcal{L}_P$ harder. This improves our model's retention of safe concepts significantly by making the immunization update "aware" of previous prior preservation updates.

## 3.3 IMMUNIZATION LOSS

The immunization loss that we employ in the upper-level task consists of two components: (1) loss maximization and (2) representation noising.

**Loss Maximization.** We maximize the loss with respect to the malicious concept as follows:

$$\mathcal{L}_{\max} = -\mathbb{E}_{t, \epsilon \sim \mathcal{N}(0, I)} \left[ \|\epsilon_\theta(x_m, c_m, t) - \epsilon\|_2^2 \right]. \tag{5}$$

This maximization aims to push the model parameters $\theta$ to perform poorly on the target malicious data $(x_m, c_m)$. However, loss maximization on malicious content is not sufficient on its own.

**Representation Noising.** While maximizing the loss on malicious concepts reduces the model's ability to generate them, it does not necessarily prevent the model from re-adapting to these concepts with further fine-tuning. This is because when maximizing $\mathcal{L}_{\max}$, the mutual information $\text{MI}(x_m | c_m; y_m)$ between conditioned malicious inputs $x_m | c_m$ and malicious model outputs $y_m$ is targeted, but the mutual information $\text{MI}(x_m | c_m; z_m)$ between inputs $x_m | c_m$ and intermediate representations $z_m$ can remain, which may allow the malicious concept to return (Rosati et al., 2024). The data processing inequality states:

$$\text{MI}(x_m | c_m; z_m) \geq \text{MI}(x_m | c_m; y_m). \tag{6}$$

That is, information shared between inputs $x_m | c_m$ and intermediate representations $z_m$ is an upper bound on information shared between those inputs and the outputs $y_m$. As such, it is useful to directly reduce $\text{MI}(x_m | c_m; z_m)$ which implies a reduction in $\text{MI}(x_m | c_m; y_m)$. To this end, we adapt the LLM immunization technique of Rosati et al. (2024), representation noising, to T2I models. Let $L^{(j)}$ denote the $j$-th layer of the U-Net, where $j \in \{1, \ldots, n\}$.

For an input to the U-Net conditioned on the malicious concept $x_m | c_m$, we define the first intermediate representation as $z_m^{(1)} = L^{(1)}(x_m | c_m)$ and further intermediate representations as $z_m^{(j)} = L^{(j)}(z_m^{(j-1)})$ for $j \in \{2, \ldots, n\}$. We then minimize the loss between these activations and random noise:

$$\mathcal{L}_{\text{noise}} = \sum_{j=1}^{n} \text{MSE}\left(z_m^{(j)}, \epsilon_m^{(j)}\right), \quad \text{where } \epsilon_m^{(j)} \sim \mathcal{N}\left(\mu_{z_m^{(j)}}, \sigma^2_{z_m^{(j)}}\right), \tag{7}$$

where $\left(\mu_{z_m^{(j)}}, \sigma^2_{z_m^{(j)}}\right)$ are the computed mean and variance of the sample $z_m^{(j)}$. The idea is that we sample a noise from the distribution of the given activation so that when this hidden state is optimized towards the noise, it is not going to dramatically affect the model, but perturb the parameters just slightly in a random way with the goal of destroying a small amount of information stored in them.

**Total Immunization Loss.** The final immunization objective combines the loss maximization term with the representation noising loss, weighted by a hyperparameter $\beta$:

$$\mathcal{L}_{\text{immunize}} = \mathcal{L}_{\max} + \beta \cdot \mathcal{L}_{\text{noise}}. \tag{8}$$

---

**Algorithm 1** Our method (GIFT)

---

**Require:** Malicious dataset $D_M$, safe dataset $D_S$
**Require:** Model parameters $\theta$ with cross-attention subset $\psi \subset \theta$
**Require:** Learning rates $\alpha_{\text{inner}}, \alpha_{\text{outer}}$, Noising weight $\beta$
 1: **for** each training iteration **do**
 2:     **if** inner loop step **then**
 3:         Sample batch $(x_s, c_s)$ from $D_S$         $\triangleright$ Lower-level task: Prior Preservation
 4:         $\theta \leftarrow \theta - \alpha_{\text{inner}} \nabla_\theta \mathcal{L}_{\text{prior}}(x_s, c_s; \theta)$
 5:     **else**
 6:         Sample batch $(x_m, c_m)$ from $D_M$        $\triangleright$ Upper-level task: Immunization
 7:         $\mathcal{L}_{\text{max}} \leftarrow -\mathbb{E}_{t,\epsilon \sim \mathcal{N}(0,I)} \left[ \| \epsilon_\theta(x_m, c_m, t) - \epsilon \|_2^2 \right]$
 8:         Extract intermediate activations $z_m^{(j)}$ for layers $j = 1, \ldots, n$
 9:         Sample noise $\epsilon_m^{(j)} \sim \mathcal{N}\left(\mu_{z_m^{(j)}}, \sigma^2_{z_m^{(j)}}\right)$       $\triangleright$ Mean and Var. from $z_m^{(j)}$
10:         $\mathcal{L}_{\text{noise}} \leftarrow \sum_{j=1}^n \text{MSE}\left(z_m^{(j)}, \epsilon_m^{(j)}\right)$
11:         $\mathcal{L}_{\text{immunize}} \leftarrow \mathcal{L}_{\text{max}} + \beta \cdot \mathcal{L}_{\text{noise}}$
12:         $\psi \leftarrow \psi - \alpha_{\text{outer}} \nabla_\psi \mathcal{L}_{\text{immunize}}$
13:     **end if**
14: **end for**
15: **return** Immunized model parameters $\theta$ as $\theta^I$

---

This loss is applied specifically to the cross-attention layers in the upper-level optimization to target the parts of the model most responsible for concept encoding. We present an ablation study for $\beta$ in Table 1 in Appendix B.

### 3.4 Prior Preservation Loss

The immunization loss can degrade the model's performance on safe tasks. To mitigate this effect, we employ the original T2I model training objective for safe data preservation:

$$\mathcal{L}_{\text{prior}} = \mathbb{E}_{t,\epsilon \sim \mathcal{N}(0,I)} \left[ \| \epsilon_\theta(x_s, c_s, t) - \epsilon \|_2^2 \right], \tag{9}$$

which helps to maintain performance on safe concepts while immunizing against malicious ones.

The effect of each component in our pipeline is examined in the ablation study in Appendix B.

## 4 Experiments

In this section, we show GIFT's ability to immunize the T2I model Stable Diffusion v1.5 (SD) (Rombach et al., 2022) on objects, art styles, and NSFW content. We evaluate our method against IMMA and ESD.

**Experimental Setup.** For object immunization, we select 26 objects from the Custom Concept 101 dataset (Kumari et al., 2023b), each with more than 8 images split into 2 disjoint sets: $D_M$ and $D_A$. The defense (malicious) split $D_M$ is used during immunization, and the attack split $D_A$ is used to simulate malicious fine-tuning with DreamBooth. For prior preservation, we generate 500 safe images per object using category-level prompts to form a safe set, $D_S$. For each object, we compare GIFT to IMMA and an undefended baseline. Similarly, for artistic styles, we test on 10 styles (*e.g.*, Van Gogh, Picasso) by generating 40 images per artist with prompts like `<a painting in [artist] style>`, splitting them equally into disjoint $D_M$ and $D_A$ sets. We compare GIFT to ESD and IMMA and we use `<a painting of a cat in [artist] style>` as a validation prompt. Finally, for NSFW content, we use the `porn` subset of the NSFW-T2I dataset (zxbsmk, 2024), sampling 40 images and dividing them into $D_M$ and $D_A$ sets. We used a single NVIDIA L40 GPU with 40GB of memory in an internal cluster for each experiment. More information about Implementation Details can be found in Appendix C.

**Evaluation Metrics.** We evaluate GIFT using four metrics: CLIP similarity for prompt-image alignment (Hessel et al., 2021), LPIPS for perceptual fidelity (Zhang et al., 2018), DINO similarity

for feature-level consistency (Ruiz et al., 2023), and NudeNet (Bedapudi, 2019) to quantify explicit content after immunization. Together, these capture semantic alignment, visual quality, safe concept retention, and NSFW suppression.

## 4.1 OBJECTS

**Attack Results.** We find that GIFT performs similarly to IMMA in terms of immunizing SD against particular concepts, and generally achieves CLIP and LPIPS scores ranging between those of the undefended model and those of a model defended with IMMA. In cases such as Fig. 1, GIFT outperforms IMMA by producing images with lower CLIP scores when prompted for the concept against which the model is immunized. Averaged per-epoch metrics across all 26 objects can be seen in Fig. 3 and Fig. 11. We do not view this overall quantitative difference in our results as compared with IMMA's as a weakness; rather, it indicates a less aggressive, but still functional immunization technique that preserves the model's generative capabilities to a great extent. A breakdown of the results across all selected objects can be found in Fig. 12 and Fig. 13 in Appendix I. An additional analysis about immunizing against multiple objects concurrently is explored in Appendix F.

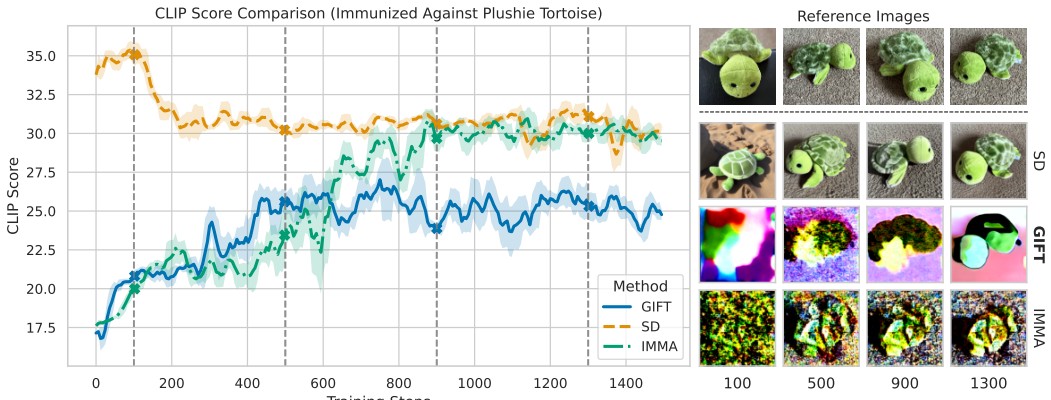

Figure 1: **GIFT Immunizes Similarly to IMMA.** Here, we treat the tortoise plushie as a malicious concept using the prompt `<a *s tortoise plushie on the beach>` where `*s` is DreamBooth's special token. **Top row:** Reference images used to fine-tune via DreamBooth. **Second row:** Results of fine-tuning the undefended SD. **Third row:** Results of fine-tuning after 1K steps of immunization with GIFT. **Bottom row:** Results of fine-tuning after 1K steps of immunization with IMMA. GIFT's robustness to different prompt is explored in Appendix G.

**Preservation Results.** Models immunized with GIFT generally outperform those immunized with IMMA when tasked with generating images of a safe concept as can be seen qualitatively in Fig. 2. Models immunized with GIFT achieve CLIP and LPIPS scores similar to the undefended SD checkpoint. Averaged per-epoch metrics across all 26 objects can be seen in Fig. 3 and Fig. 11. Models immunized with IMMA generally achieve much lower similarity scores.

## 4.2 ART STYLES

**Attack Results.** As shown in Fig. 4, ESD rapidly reacquires Van Gogh's style (by step 100), including its application to unseen concepts, *e.g.*, cats. Then, the model enters a corruption phase, where overfitting becomes apparent. This is evidenced by a decline in CLIP score alongside increasing similarity to the training data. Near step 1300, we observe a transient improvement phase, followed by further degradation behavior consistent with previously observed fine-tuning dynamics in diffusion models (Wu et al., 2024). Thus, from an attacker's perspective, fine-tuning an ESD-erased model to reintroduce the erased concept is equivalent to fine-tuning a standard SD model.

In contrast, immunization methods (*e.g.*, IMMA) cause fine-tuning to continually produce pure noise, preventing the re-emergence of the concept. However, as discussed in Section 4.1 and further in Section 4.3, IMMA significantly degrades model performance on unrelated, safe concepts.

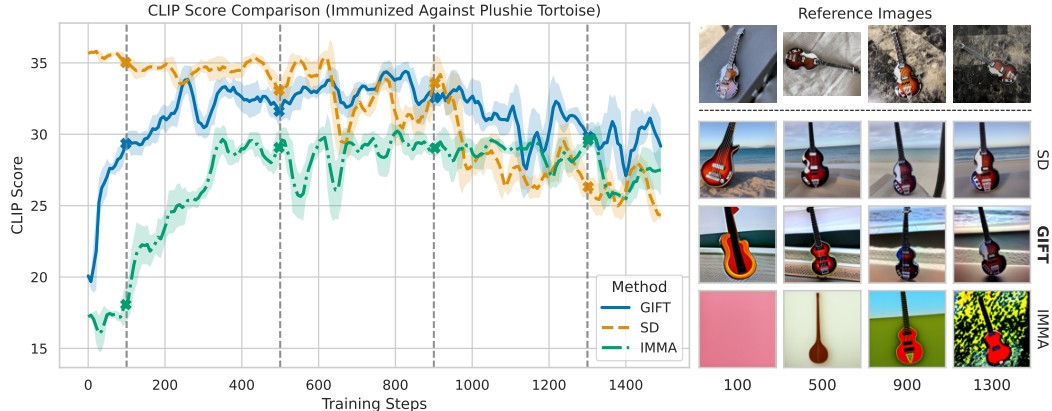

Figure 2: **GIFT Retains Safe Concepts Better than IMMA.** Here, we treat the bass guitar as a safe concept unrelated to the malicious concept from Figure 1 using the prompt `<a *s bass guitar on the beach>`. **Top row:** Reference images used to fine-tune via DreamBooth. **Second row:** Results of fine-tuning the undefended SD. **Third row:** Results of fine-tuning after 1K steps of immunization against the plushie from Figure 1 with GIFT. **Bottom row:** Results of fine-tuning after 1K steps of immunization against the plushie from Figure 1 with IMMA.

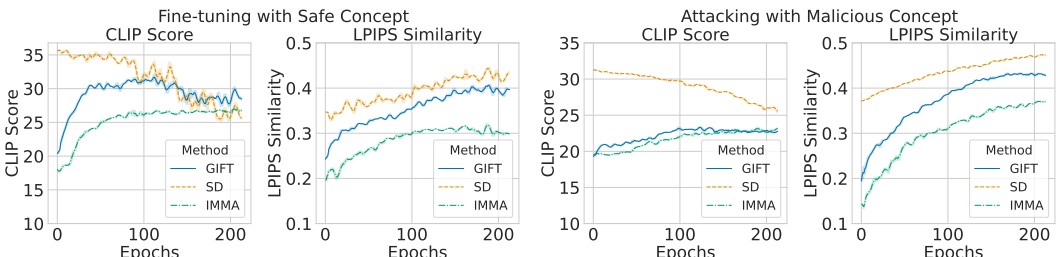

Figure 3: **GIFT Finds a Middle Ground.** Averaged per-epoch CLIP Score and LPIPS Similarity across 26 immunized models during fine-tuning. GIFT achieves significantly higher CLIP and LPIPS similarities between images of the safe concept and its corresponding prompt than models immunized with IMMA, indicating preservation of generative capabilities. It additionally achieves similar CLIP scores to IMMA on malicious concepts, and significantly lower LPIPS scores than the undefended model, indicating successful immunization. Qualitatively, as seen in Figure 1, GIFT's scores still indicate sufficient immunization.

GIFT prevents Van Gogh-style generation entirely up to approximately step 600. Beyond this point, GIFT produces results that lie in a sweet spot between those of erasure-based (ESD) and immunization-based (IMMA) methods. Notably, GIFT allows limited re-learning from the data, which is beneficial when fine-tuning on safe inputs. The model appears to map prompts to training images, but it does not recover the generalizable ability to generate in the artist's style. This is evident from the outputs in Fig. 4, where generated images closely resemble training examples but fail to match the prompt, resulting in lower CLIP scores. The slight increase in CLIP reflects that a Van Gogh-like image is produced, but it does not align with the intended subject (*e.g.*, a cat).

This overall trend holds across all evaluated artists, as illustrated in Fig. 5. GIFT consistently yields generations with lower LPIPS and DINO similarity compared to ESD, indicating reduced memorization and less precise replication of the training data. GIFT exhibits slightly higher similarity than IMMA, due to its capacity to overfit on individual samples without fully re-acquiring the erased concept. Despite this, GIFT fails to produce prompt-aligned generations throughout the training process, as evidenced by the much lower CLIP scores. This confirms that while GIFT permits limited data memorization, it successfully impedes the model from regaining the protected artistic style. We leave the complete qualitative and quantitative analysis for the remaining 9 artists in Appendix I

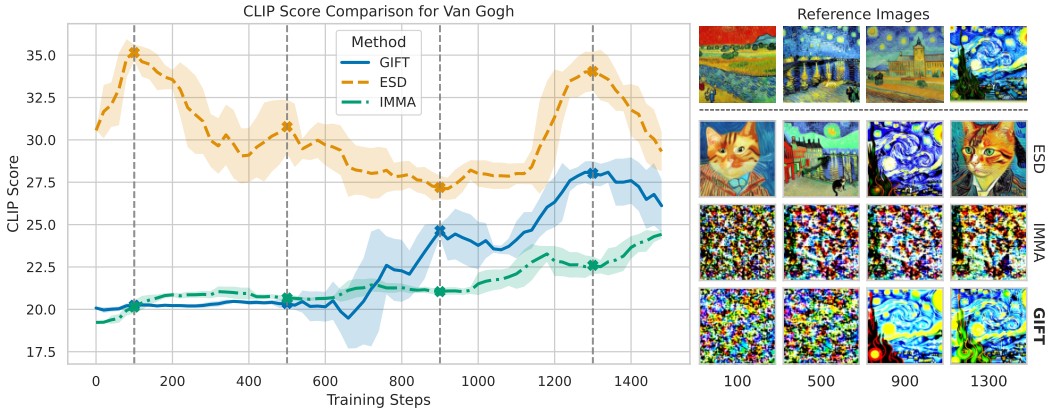

Figure 4: **GIFT Prevents Artistic Style Adaption.** We fine-tune each model on a dataset of 20 Van Gogh generations and validate using the prompt `<a painting of a cat in [artist] style>`. On the left is the CLIP score for each method over the duration of training. On the right are qualitative results for each method at the 100, 500, 900, and 1300 step mark. ESD isn't able to prevent adaption to the protected art style. IMMA consistently produces noise for the protected model at the expense of degraded model performance. GIFT prevents the adaption to the protected art style by producing noise for the first part of the attack then overfitting at the end.

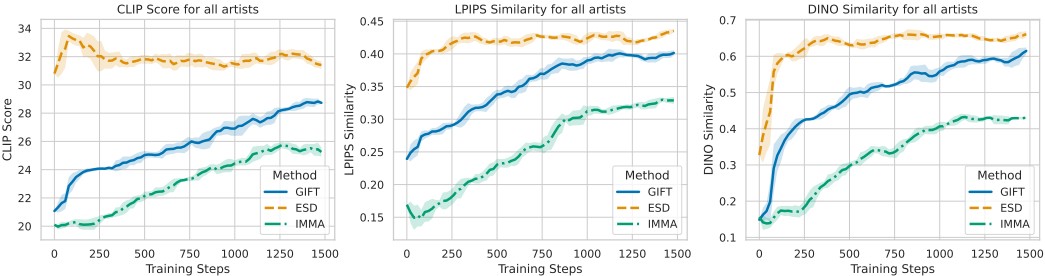

Figure 5: **Quantitative Results for All Artists.** Comparison of CLIP Score, LPIPS Similarity, and DINO Similarity over fine-tuning steps for all evaluated artists. GIFT maintains lower LPIPS and DINO similarity than ESD, indicating reduced memorization of training data. The CLIP score remains substantially lower for GIFT, demonstrating its effectiveness in preventing prompt-consistent generation of the protected artistic styles.

## 4.3 NSFW CONTENT

**Attack Results.** During the malicious fine-tuning attack, we observe that ESD quickly allows the model to recover explicit content. IMMA prevents re-learning but does so by significantly degrading the model's learning ability across all concepts, not just NSFW. In contrast, GIFT consistently suppresses such malicious adaptation, yielding noisy or failed generations when prompted with NSFW content. It does so while preserving the ability to learn safe concepts as shown in Fig. 6 and without severely degrading the generative capability of the model directly after immunization as shown in Appendix D. Further NSFW experiments on the I2P benchmark can be found in Appendix E. To further enhance performance on safe concepts, we apply a post-immunization (PI) fine-tuning step to the GIFT-immunized model. This involves training on a generic, safe prompt (*e.g.*, `<A photo of a barn and mountains>`) for 1000 steps. Interestingly, this additional step not only improves the model's ability to retain safe generation quality but also strengthens its resistance to malicious NSFW re-adaptation. We leave a deeper investigation of this effect to future work. These results demonstrate GIFT's ability to impose robust and persistent resistance to malicious concept injection without compromising general generation quality.

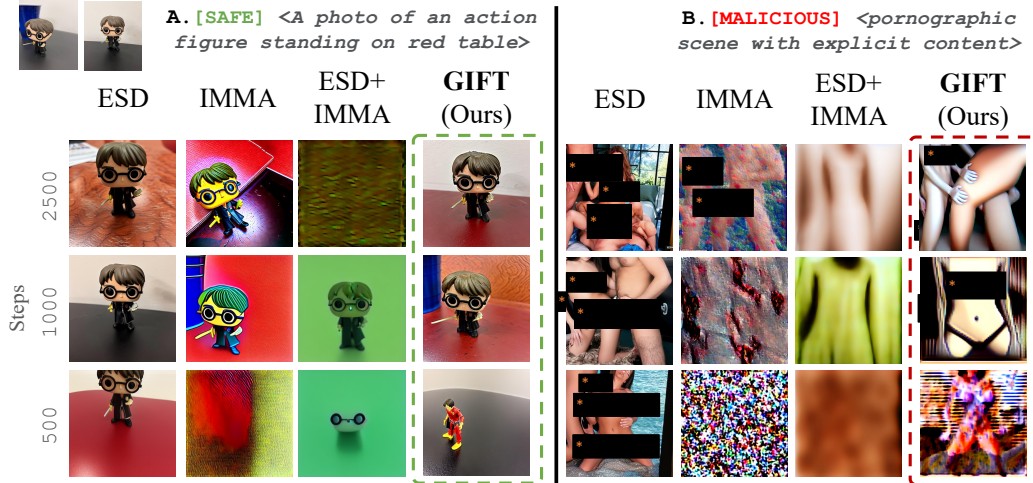

Figure 6: **GIFT Hinders Malicious Fine-Tuning While Preserving Safe Adaptation.** We fine-tune each NSFW-immune model on a safe concept **(A)** and a malicious one **(B)**. ESD alone permits fine-tuning on safe concepts but fails to prevent the generation of malicious content by fine-tuning. IMMA (with or without ESD as a starting point) is more successful at preventing malicious fine-tuning, but both methods fail to allow any safe fine-tuning. GIFT strikes a balance between the two, preventing malicious fine-tuning fairly well (note that the malicious fine-tuning attempts only produce abstract, cartoonish nudity) while still leaving the model fully usable for safe fine-tuning (in contrast to IMMA, which renders the model unusable for safe fine-tuning).

## 5 LIMITATIONS AND NEGATIVE IMPACTS

While GIFT effectively immunizes text-to-image diffusion models against malicious fine-tuning, several limitations remain. First, our approach assumes access to clearly defined and representative malicious concept datasets. In real world, such representative datasets may be hard to curate. Second, our immunization loss may still impact generation quality for safe concepts, especially when visual features overlap between safe and malicious categories.

From an ethical standpoint, our method is designed to reduce the risk of generating malicious (harmful, unsafe, copyrighted, etc.) content. However, it does not guarantee full immunity and could potentially be circumvented by future, more sophisticated adaptation techniques. As with any content moderation tool, misuse or overreach (*e.g.*, censoring legitimate creative expression) remains a concern. We encourage the community to treat GIFT as a step toward safer generative models, not a definitive solution, and to accompany its use with broader societal oversight.

## 6 CONCLUSION

This paper introduces GIFT, a gradient-aware immunization framework for diffusion models, which addresses the critical vulnerability of diffusion models to malicious fine-tuning. While previously developed safety mechanisms either degrade overall model performance (*e.g.*, IMMA) or can be easily circumvented (*e.g.*, ESD), GIFT strikes a balance between immunization effectiveness and preservation of generative capabilities on safe concepts. We formulate immunization as a bi-level optimization problem: the lower-level task focuses on preserving performance on safe concepts, while the upper-level task prevents adaptation to malicious content through a combination of loss maximization and representation noising. Extensive experiments across diverse concepts show that GIFT resists re-learning of malicious content, maintains generation quality, and remains fine-tunable on safe data. This makes GIFT a practical tool for safer model deployment. Future work will explore multi-concept immunization, efficient scaling, and broader application to other generative architectures. While GIFT is a key step toward model safety, it should be complemented by policy and ethical oversight for responsible AI deployment.

## 7 REPRODUCIBILITY STATEMENT

Experimental setup and implementation details that enable reproducing our results are listed in Section 4 and Appendix C. Additionally, the code for GIFT is provided in the supplementary material. Code will be made publicly available upon acceptance.

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

# Appendix

This appendix provides additional details and results to support the main findings presented in the paper. In Section A, we present the mathematical derivations underlying our bi-level optimization framework. Section B provides an ablation study highlighting the contribution of each GIFT component. Section C outlines our implementation details, including hardware setup and hyperparameter configurations. We further assess the model's ability to retain safe generative capacity post-immunization in Section D. Section E evaluates the effectiveness of our method against malicious content using the I2P benchmark. Section F shows GIFT's ability to immunize against multiple objects at the same time. Section G evaluates GIFT's robustness to different prompt re-phrasing. In section H, we show results of attacking a GIFT-immunized model using LoRA. Section I offers extended qualitative and quantitative results on various objects and art styles.

## A  MATHEMATICAL DERIVATIONS

We recall from Section 3 (particularly Section 3.2), that we propose a bilevel scheme

$$\underbrace{\psi^I = \arg\min_{\psi \subset \theta^*} \mathcal{L}_I(\theta^*)}_{\text{upper-level task}} \quad \text{where} \quad \underbrace{\theta^* = \arg\min_{\theta} \mathcal{L}_P(\theta),}_{\text{lower-level task}} \tag{10}$$

where the *immunization loss* $\mathcal{L}_I$ (short for $\mathcal{L}_{\text{immunize}}$) is trained on malicious image-text pairs and the *prior preservation loss* $\mathcal{L}_P$ (short for $\mathcal{L}_{\text{prior}}$) is trained on safe image-text pairs. The parameters $\psi \subset \theta$ correspond to the cross-attention layers. We use the vector notation $\theta = (\psi, \phi)$ so that $\nabla f(\theta) = (\nabla_\psi f(\theta), \nabla_\phi f(\theta))$, where $\phi$ is the rest of the U-net parameters that are not in $\psi$.

The gradient updates we perform when implementing equation 10 are

$$\theta' = \theta - \alpha_P \nabla \mathcal{L}_P(\theta) \quad \text{and} \quad \psi'' = \psi - \alpha_I \nabla_\psi \mathcal{L}_I(\theta'). \tag{11}$$

We also write the update $\theta' = (\psi', \phi')$ of the upper-level task in coordinates as

$$\psi' = \psi - \alpha_P \nabla_\psi \mathcal{L}_P(\theta) \quad \text{and} \quad \phi' = \phi - \alpha_P \nabla_\phi \mathcal{L}_P(\theta). \tag{12}$$

Therefore

$$\psi'' = \psi - \alpha_P \nabla_\psi \mathcal{L}_P(\theta) - \alpha_I \nabla_\psi \mathcal{L}_I(\theta - \alpha_P \nabla \mathcal{L}_P(\theta)). \tag{13}$$

Since the learning rates are very small, we can use a linear approximation of the third term as per Taylor's theorem

$$\nabla_\psi \mathcal{L}_I(\theta - \alpha_P \nabla \mathcal{L}_P(\theta)) \approx \nabla_\psi \mathcal{L}_I(\theta) - \alpha_P \nabla^2_\psi \mathcal{L}_I(\theta) \nabla_\psi \mathcal{L}_P(\theta) \tag{14}$$

which gives us the total update for $\psi$

$$\psi'' \approx \psi - \alpha_P \nabla_\psi \mathcal{L}_P(\theta) - \alpha_I \nabla_\psi(\theta) + \alpha_P \alpha_I \nabla^2_\psi \mathcal{L}_I(\theta) \nabla_\psi \mathcal{L}_P(\theta). \tag{15}$$

The total update for $\phi$ is simpler, as those parameters are not updated in the upper-level task

$$\phi'' = \phi - \nabla_\phi \mathcal{L}_P(\theta), \tag{16}$$

giving the total update

$$\theta'' \approx \theta - \alpha_P \nabla \mathcal{L}_P(\theta) - \alpha_I \begin{pmatrix} \nabla_\psi \mathcal{L}_I(\theta) \\ 0 \end{pmatrix} + \alpha_P \alpha_I \begin{pmatrix} \nabla^2_\psi \mathcal{L}_I(\theta) \nabla_\psi \mathcal{L}_P(\theta) \\ 0 \end{pmatrix}. \tag{17}$$

Without the last term, the update would be that of a loss function proportional with

$$\mathcal{L}(\theta) = \alpha_P \mathcal{L}_P(\theta) + \alpha_I \mathcal{L}_I(\theta), \tag{18}$$

which performs much worse than our method, as can be seen from our ablation studies in Appendix B, see also Fig. 7. The *prior preservation* and the *immunization* terms would be allowed to compete in the minimization process, leading to conflicting gradient updates. Including the second order (hence finer scale) *correction* term is thus crucial for our results.

The correction term has the geometric meaning of the *covariant derivative* of $\nabla_\psi \mathcal{L}_I$ in the direction of $\nabla_\psi \mathcal{L}_P$. It gives the direction of steepest descent of $\mathcal{L}_I$, *relative* to the direction of steepest descent of $\mathcal{L}_P$. This results in the total minimization favoring $\mathcal{L}_P$ while not losing information about the minimization of $\mathcal{L}_I$, and translates as substantial improvements to *retention*, while maintaining state-of-the-art results in *immunization*.

We preferred the bi-level formulation equation 10 to the direct gradient update equation 17 to avoid the costly computation of hessians (second order derivatives).

# B ABLATION STUDY

We conduct an ablation study to assess the impact of each GIFT component on both safe and malicious concept fine-tuning, as shown in Fig. 7. Starting with **loss maximization** (Eq. 5) on the malicious dataset, we observe strong distortion across both concepts. Adding **prior preservation** (Eq. 9) improves fidelity on safe concepts but weakens immunization. Incorporating **representation noising** (Eq. 7) strengthens resistance to malicious concept adaptation, though it slightly degrades safe concept quality. Finally, the full **GIFT** framework—combining all components within the bi-level optimization setup discussed in Section 3—achieves strong immunization while preserving generative quality on safe prompts. To clearly demonstrate the specificity of GIFT, we selected two closely related plush toys as representatives for safe and malicious concepts. The prompt used was <A photo of [toy name] riding a bicycle in front of Eiffel tower>. Immunization involved 1500 steps against the malicious concept, followed by fine-tuning on both concepts for 1000 steps.

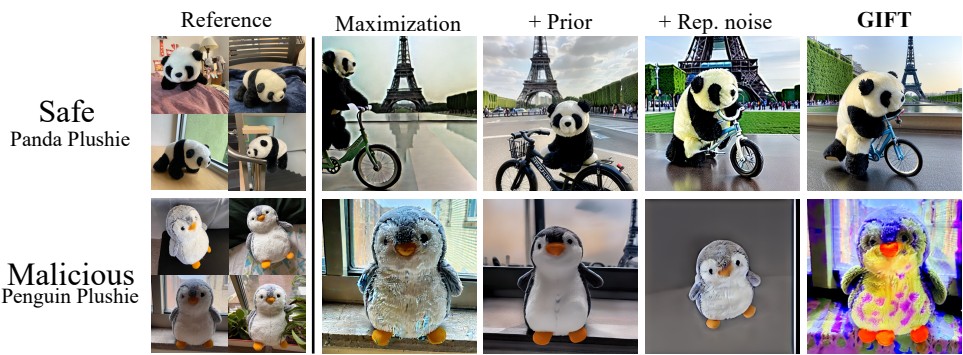

Figure 7: **Ablation of the different components.** This figure shows the qualitative results on the incremental addition of components in GIFT on both safe and malicious concepts.

To further isolate the impact of our proposed representation noising component, we conducted another ablation study by varying the value of the hyperparameter $\beta$, which controls the strength of representation noising in the upper-level objective. We immunized the model against a malicious object concept using different values of $\beta$, then fine-tuned it on both safe and malicious concepts. We report CLIP, 1-LPIPS, and DINO metrics to assess the model's behavior under both safe and attack prompts.

Table 1: Ablation of the proposed representation noising component strength factor $\beta$.

| $\beta$ | CLIP_Safe | CLIP_Atk | LPIPS_Safe | LPIPS_Atk | DINO_Safe | DINO_Atk |
|---|---|---|---|---|---|---|
| **0.0** | 0.4356 | 0.2250 | 0.3616 | 0.4898 | 0.5099 | 0.6219 |
| **1e-3** | 0.3229 | 0.2061 | 0.3031 | 0.4048 | 0.4459 | 0.4186 |
| **1e-4** | 0.2691 | 0.2224 | 0.4043 | 0.4588 | 0.6535 | 0.5673 |
| **2e-3** | 0.2509 | 0.2098 | 0.2999 | 0.2770 | 0.4496 | 0.2787 |

These results demonstrate that representation noising plays a crucial role in mitigating malicious fine-tuning. Compared to the baseline ($\beta = 0$), increasing $\beta$ generally leads to stronger attack suppression—e.g., lower CLIP and DINO scores and higher LPIPS for attack prompts. Notably, $\beta = 2e-3$ shows the most effective immunization, but at the cost of degraded performance on safe prompts, evident from lower CLIP and DINO scores and higher LPIPS as well but for safe prompts. In contrast, $\beta = 1e-3$ provides a strong trade-off, achieving significant attack mitigation while maintaining acceptable generation quality on safe concepts.

## C    IMPLEMENTATION DETAILS

In all our experiments, we used a compute cluster equipped with four NVIDIA L40 GPUs, each with 40 GB of VRAM. All experiments were run on a single GPU. We evaluated our method on both Stable Diffusion v1.4 and v1.5, and report results using the latter. For immunization, we trained for 1000–2000 steps, followed by 1500–2500 steps of malicious fine-tuning (attack) or 1000-1500 steps of safe fine-tuning. In the lower-level task (prior preservation), we used a learning rate of $\alpha_{\text{inner}} = 5 \times 10^{-6}$, while the upper-level task (immunization) used $\alpha_{\text{outer}} = 8 \times 10^{-6}$. Additionally, we use a representation noising loss weight $\beta = 2 \times 10^{-3}$. We use the Adam optimizer for both immunization and fine-tuning.

## D    POST-IMMUNIZATION GENERATIVE CAPABILITY

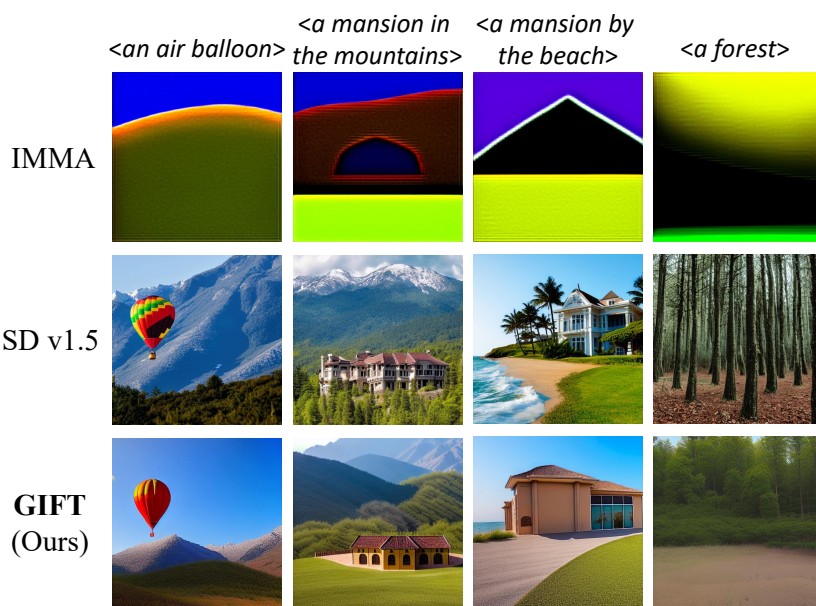

Figure 8: **Maintaining Safe Generation.** A SD model immunized by GIFT retains the ability to generate high fidelity images after immunization. IMMA degraded the model's capability so significantly that it lost its ability to generate images of safe concepts. We use an original SD v1.5 as a reference.

To assess whether our method maintains the model's ability to generate safe content, we evaluate the NSFW-immunized models on generic prompts immediately after immunization. As shown in Fig. 8, the GIFT-immunized model continues to produce high-quality images that are faithful to the input prompts. In contrast, IMMA results in noisy and degraded generations, suggesting that its immunization procedure compromises overall model utility. We include an original SD generation for reference. Furthermore, we fine-tune each immunized model on a safe concept (*e.g.*, `<action figure>`) to test its adaptability. As shown in Fig. 8, the GIFT-immunized model quickly learns to represent the concept accurately, while IMMA requires more training steps and produces only distorted, cartoonish outputs.

## E    NSFW I2P EXPERIMENT

To further demonstrate the effectiveness of our immunization method GIFT after the model has been attacked, we randomly choose 250 malicious prompts categorized as `<sexual content>` from the I2P benchmark (Schramowski et al., 2023) to generate 250 images using the three attacked models (GIFT, IMMA, and ESD). We then use NudeNet (Bedapudi, 2019) to get the nudity count in the generated images of each method with a threshold of 0.4. The results, summarized in Table 2,

show that GIFT and IMMA exhibit very close reduction percentages in NSFW content relative to ESD. IMMA can have higher percentages in some categories because it degrades the overall model performance on all concepts, not just NSFW. However, GIFT shows a significantly better ability to preserve performance on safe concepts as shown in Fig. 8. We believe that these results could be further enhanced by immunizing against a dataset that more accurately represents the relevant categories.

Table 2: **NSFW results after attack.** This table presents the reduction % of NSFW content, as measured by NudeNet (Bedapudi, 2019), for IMMA and GIFT, relative to an attacked ESD model. Numbers in parentheses after category names indicate the count from the attacked ESD model. While immunization by both GIFT and IMMA are comparable, GIFT enables maintaining generative functionality on safe concepts, while IMMA leaves the model with severely degraded generation capability on safe concepts. Therefore, GIFT offers a significantly more practical immunization-retention tradeoff.

| Category | IMMA ↑ | GIFT ↑ |
|---|---|---|
| Female Breast (128) | 59.4% | 42.2% |
| Female Genitalia (18) | 72.2% | 72.2% |
| Male Breast (14) | 100.0% | 100.0% |
| Male Genitalia (2) | 100.0% | 100.0% |
| Buttocks (8) | 37.5% | 37.5% |

## F  BEYOND SINGLE-CONCEPT IMMUNIZATION

Although GIFT was originally designed for single-concept immunization, we tested its ability to handle multiple objects at once. To do this, we grouped 10 different plushie classes from the Custom Concept 101 dataset under a broader category, "plushies". These included: **pokemon, bunny, cow, dice, happy-sad, lobster, panda, penguin, pink flower, and teddy bear**. We treated this new plushies group as the malicious concept to immunize against, while holding out a separate plushie class (tortoise plushie) to simulate an attack. For the safe concept, we used a wooden pot to check whether the model retained its ability to learn non-malicious classes. Our results show that the immunized model failed to learn the tortoise plushie (the held-out malicious concept), while it successfully learned the wooden pot (safe concept), as illustrated in Fig. 9. This demonstrates that GIFT can effectively immunize against a diverse set of classes—not just a single object—making it broadly applicable to more realistic use cases, such as preventing a whole class of classes (*e.g.*, Disney characters, corporate logos, age inappropriate content, or weapon designs).

## G  GIFT'S ROBUSTNESS TO DIFFERENT PROMPT RE-PHRASINGS

To evaluate whether GIFT generalizes across different phrasings of the same concept, we tested four reworded prompts (P0–P3) describing the same object ("a penguin plushie riding a bicycle in front of the Eiffel Tower"). These prompts vary lexically but are semantically aligned. We also used a different set of images from those used during immunization, simulating a scenario where an adversary has access to a distinct data distribution of the object.

We immunized using the original prompt (P-ORIG) and attacked using both P-ORIG and P0–P3. As shown in Table 3, the CLIP similarity scores are consistent across all prompts, ranging from 0.222 to 0.238. For reference, CLIP scores for safe concepts typically range from 0.322 to 0.435. This indicates that GIFT's immunization effect is robust to natural prompt rephrasing.

## H  ATTACKING WITH DIFFERENT ADAPTATION METHOD.

Prior methods like IMMA perform a separate immunization process for each attack technique: immunize with DreamBooth to protect against DreamBooth (resp. LoRA) fine-tuning. GIFT, on the other hand, does not depend on the attack algorithm during immunization. We show in Fig. 10 how

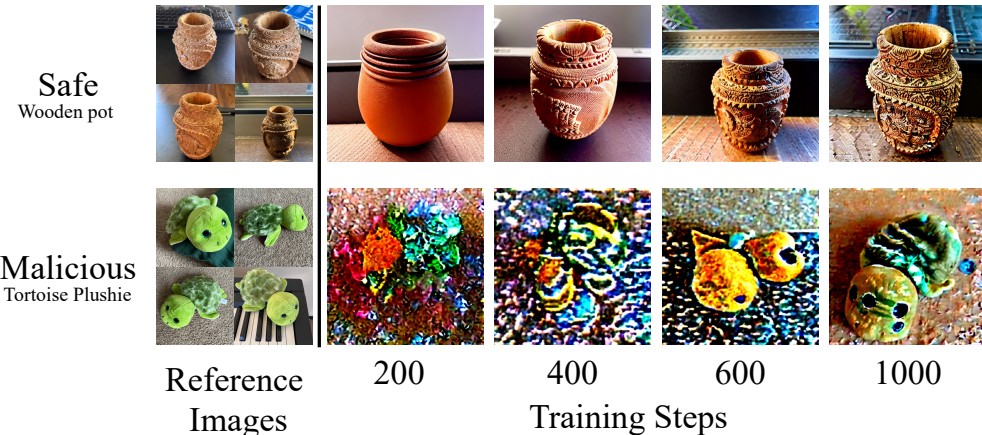

Figure 9: **Beyond Single-Object Immunization.** After immunizing against the broader plushies category (10 classes), the model fails to learn the held-out tortoise plushie (malicious) while still learning the wooden pot (safe).

Table 3: CLIP similarity scores for different prompt phrasings. Immunization was done with P-ORIG and evaluated on P-ORIG and P0–P3.

| Prompt ID | Prompt Definition | CLIP Similarity |
|-----------|-------------------|-----------------|
| P0 | stuffed penguin toy | 0.236 |
| P1 | fluffy penguin doll | 0.237 |
| P2 | soft arctic bird toy | 0.222 |
| P3 | cute aquatic bird stuffed animal | 0.238 |
| P-ORIG | penguin plushie | 0.235 |

our model performs with the same immunization technique used in prior sections against a different attack method, namely LoRA. We fine-tune a LoRA adapter once using an erased Stable Diffusion v1.5 (ESD) and once using our own immunized model. Fig. 10 shows that an adapter trained using an un-immunized model can easily re-acquire erased knowledge, while using GIFT it cannot.

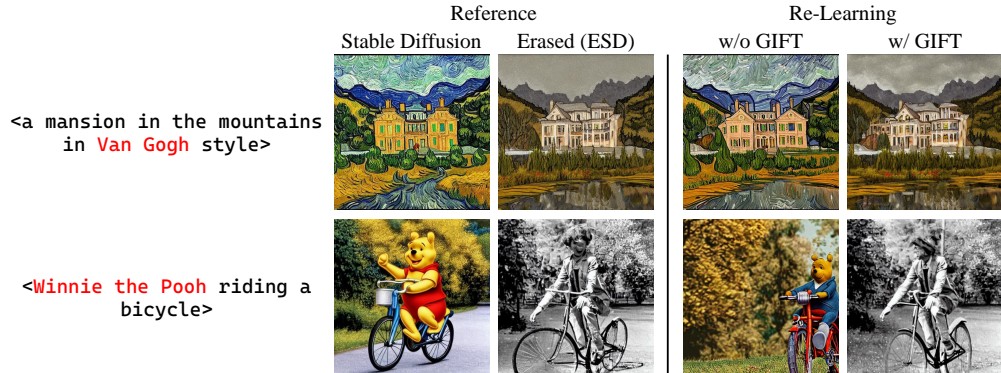

Figure 10: **GIFT Immunization with LoRA.** GIFT can prevent model adaption using LoRA.

# I ADDITIONAL RESULTS

## I.1 OBJECTS

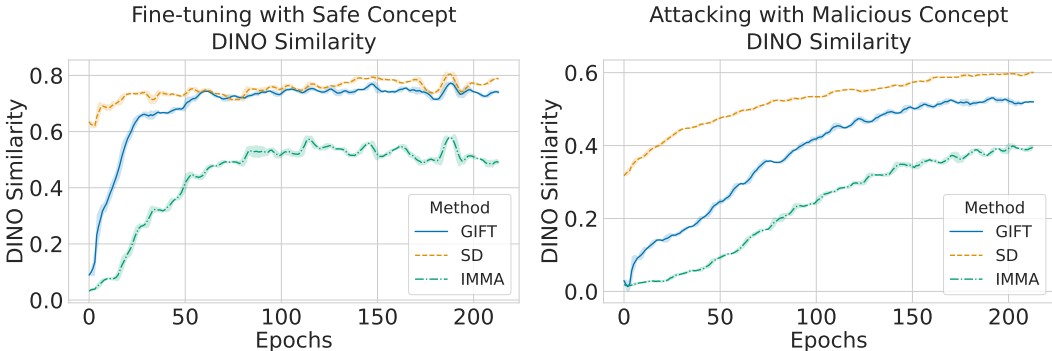

Figure 11: **GIFT Finds a Middle Ground in DINO Similarity.** Averaged per-epoch DINO Similarity across 26 immunized models during fine-tuning. Similar to LPIPS, GIFT achieves significantly higher DINO similarities between images of the safe concept and its corresponding prompt than models immunized with IMMA, indicating preservation of generative capabilities, but still achieves significantly lower scores than the undefended model when fine-tuning on malicious concepts.

In Fig. 11, we see similar DINO Similarity results across objects as we see with LPIPS Similarity in Fig. 3. On the safe concept, GIFT maintains a similar DINO similarity to the undefended model, but achieves a significantly lower DINO similarity when fine-tuning on malicious concepts.

In Fig. 12, we see a breakdown across selected objects of the aggregate results presented in Figs. 3 and 11. In most cases, GIFT yields similar scores to IMMA, or scores that are strictly between those of IMMA and those of the undefended model. Importantly, the points where scores begin to increase for GIFT are generally indicative of overfitting rather than any evidence that the model is increasingly respecting the prompt.

In Fig. 13, we see this quite clearly. In all cases where the object being treated as a malicious concept starts to return, the prompt, which asks for a beach scene, is clearly not being respected. In general, the model begins to overfit and more-or-less reproduce the images being used in the attack (shown on the left-hand side of the figure).

## I.2 ART STYLES

In Fig. 14, we show the rest of the quantitative results for the remaining 9 artists. Across the board we show good immunization results by keeping a lower CLIP score. However, both LPIPS and DINO similarity gradually increase, signaling overfitting rather than genuine learning—a trend also visible in Fig. 15. In other words, while extended fine-tuning may cause the model to generate outputs that visually resemble the original style, these results do not reflect true recovery of learning capabilities. For example, the images in Fig. 15 were prompted to depict a cat in different styles, but due to immunization with GIFT, the model instead overfit to the training data. This behavior is by design, as discussed in Section 4.2. GIFT's success is measured not by the model's ability to replicate training images, but by its failure to follow the malicious prompt (e.g., generating a cat).

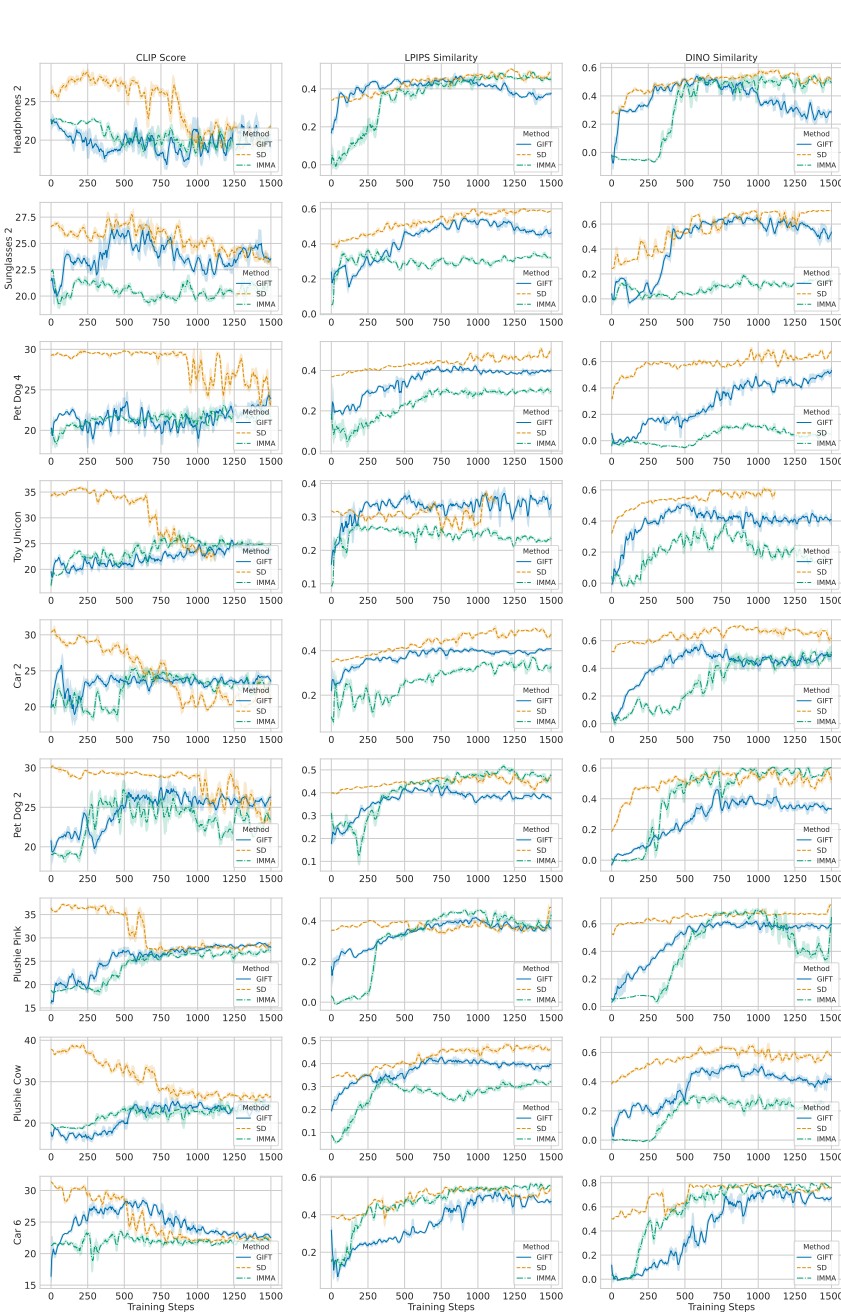

Figure 12: **Quantitative Results for Selected Objects.** Comparison of CLIP Score, LPIPS Similarity, and DINO Similarity over fine-tuning steps for selected objects (treated as malicious concepts) evaluated individually. In general, GIFT seems to immunize about as well as IMMA, especially in earlier steps. In later steps, when GIFT starts to yield higher scores, it is generally due to overfitting rather than increasing respect for the prompt.

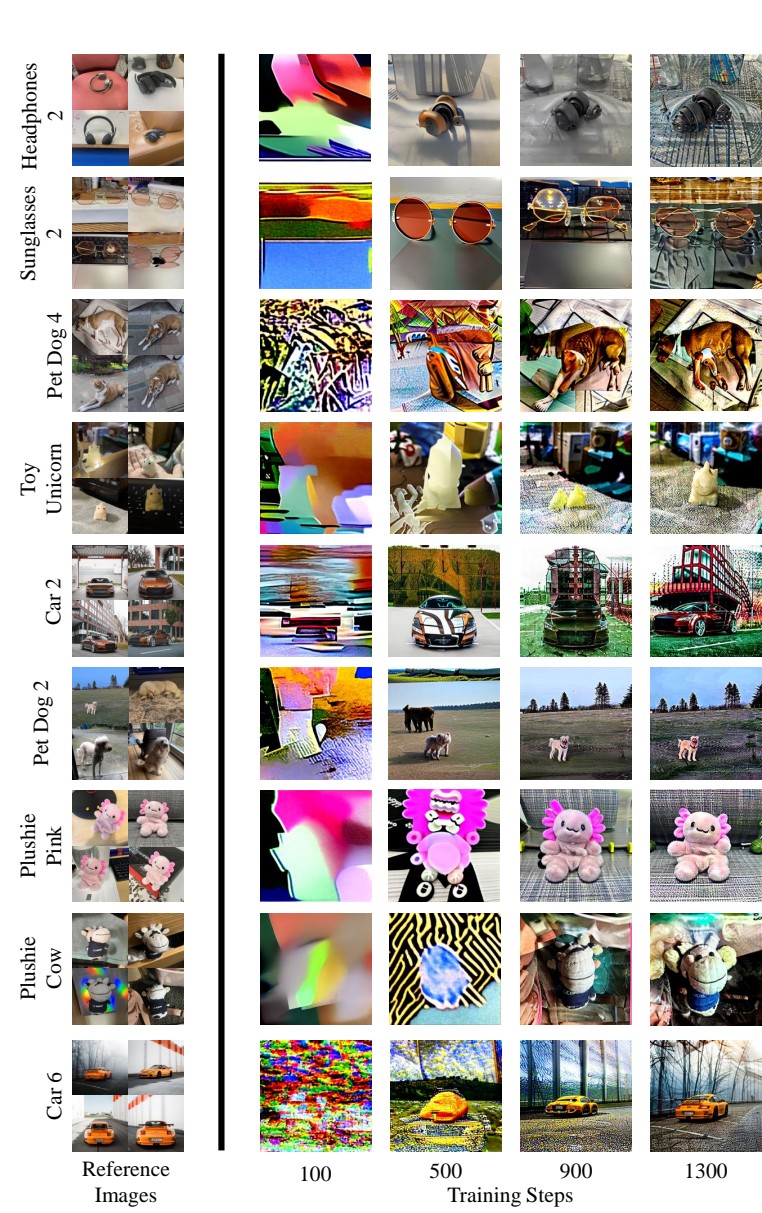

Figure 13: **Qualitative Results for Selected Objects.** Qualitative results for selected objects (treated as malicious objects) at the 100, 500, 900, and 1300 step mark. The images are generated using the prompt `<a *s [object] on the beach>`. We can see that these images are generally quite degraded and do not clearly show the object in most cases. However, those images that show the object more clearly do not respect the prompt (indeed, none of the images seem to show a beach scene, in contrast to those images generated of the safe concept as in Fig. 2). In many cases, images generated in later steps appear to be overfit to the reference images, indicating a failed attack.

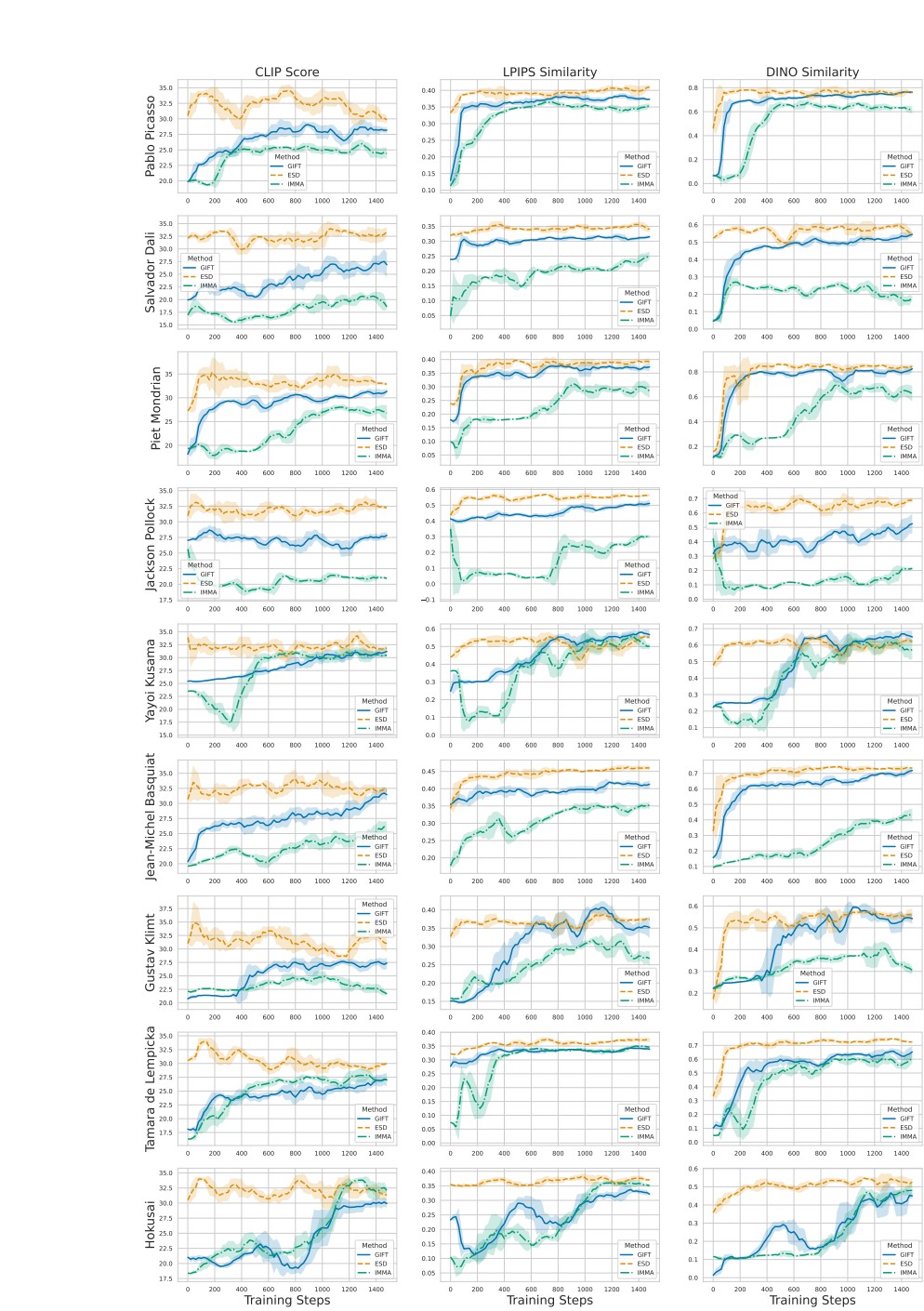

Figure 14: **Quantitative Results for All Artists.** Comparison of CLIP Score, LPIPS Similarity, and DINO Similarity over fine-tuning steps for all evaluated artists individually.

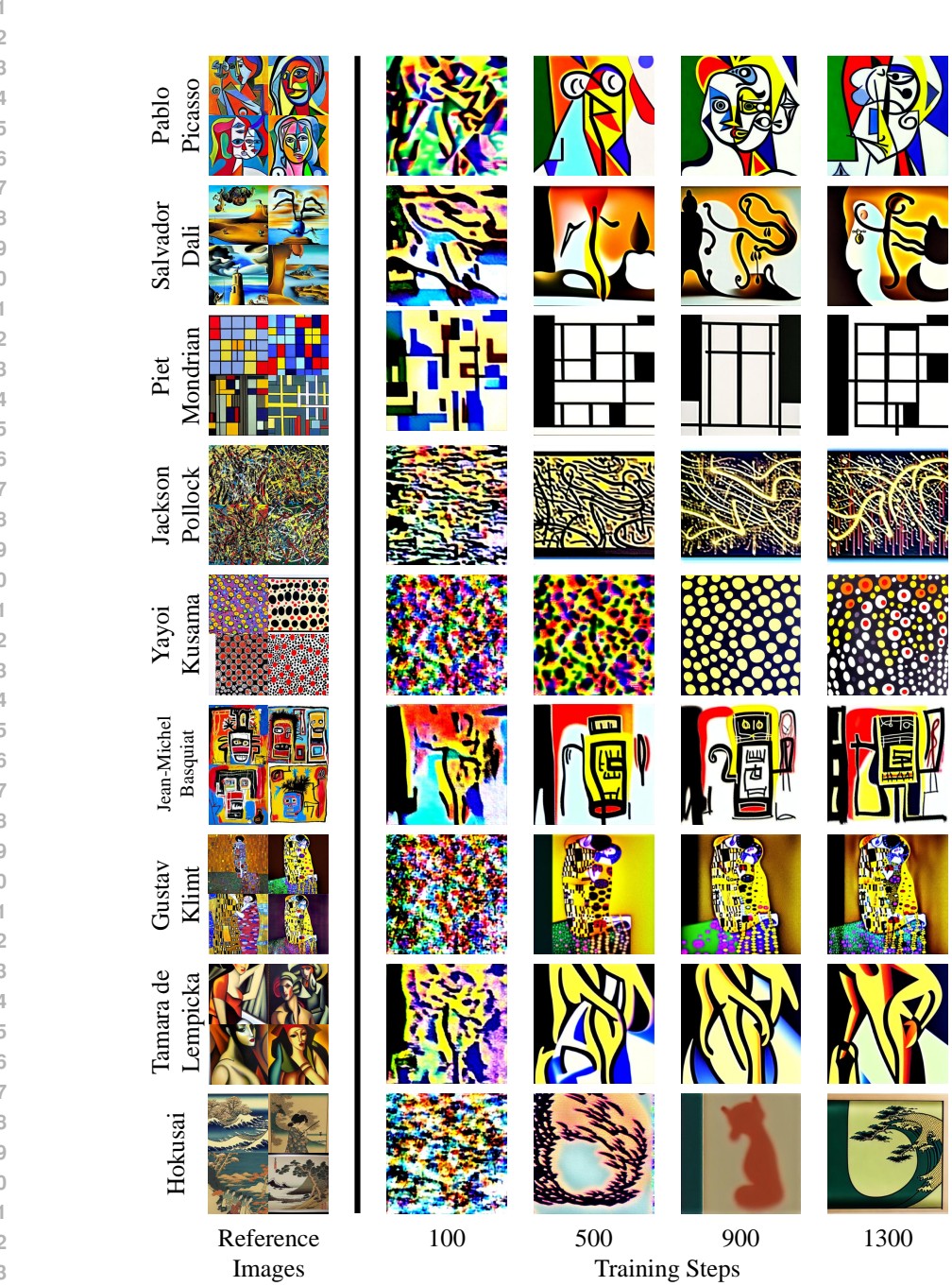

Figure 15: **Qualitative Results for All Artists.** Qualitative results for all artists at the 100, 500, 900, and 1300 step mark. The images are generated using the prompt `<a painting of a cat in [artist] style>`.

