# OpenReview forum: "GIFT: Gradient-aware Immunization of diffusion models against malicious Fine-Tuning with safe concepts retention"
_ICLR.cc/2026/Conference — ICLR 2026 Conference Withdrawn Submission_

### Official Review · Reviewer_HJV3 · 2025-10-25

**Soundness:** 2
**Presentation:** 1
**Contribution:** 2
**Rating:** 2
**Confidence:** 4

**Summary:**

This paper proposes GIFT, an immunization framework designed to make open-source diffusion models resistant to malicious fine-tuning (e.g., recovering erased or unsafe concepts) while preserving their ability to generate safe content. The method formulates the immunization process as a bi-level optimization problem, which both maintains safe generation and reduces the model’s adaptability to malicious concepts, even under adversarial fine-tuning attacks. Experiments on objects, art styles, and NSFW concepts demonstrate that GIFT effectively mitigates malicious fine-tuning while retaining strong generative capabilities for safe content.

**Strengths:**

1. Defending open-source diffusion models from adversarial fine-tuning is an important and timely problem.
2. Evaluation across multiple domains with comparisons against IMMA and ESD, demonstrating a consistent balance between safety and usability.
3. The approach offers the flexibility to suppress arbitrary concepts.

**Weaknesses:**

1. The bi-level optimization design, while well-motivated, seems mostly heuristic without a clear theoretical or empirical justification for why it outperforms a simple joint loss.
2. The method assumes access to separate malicious and safe datasets, but these are difficult to define or collect in practice. For instance, defining a “safe” concept as “any other concept” makes the safe set effectively infinite and impractical to obtain.
3. Limited practicality: The immunized model cannot generate desired outputs directly from prompts; users must fine-tune with reference images, which is neither common nor convenient. Moreover, if I understand correctly, the reference image’s concept must lie within the safe dataset distribution used during immunization, further reducing the model’s utility.

**Questions:**

Please refer to the weakness and the following:
1. Why does Maximization + Rep. Noise outperform the full GIFT (Maximization + Rep. Noise + Prior) on malicious concepts? The prior term should preserve safe content without weakening immunization. Also, why does Figure 7 show significantly weaker suppression of malicious concepts than Figure 1?

---

> ### Author Response · Authors · 2025-11-21
>
> 1. We thank the reviewer for their comments. We refer the reviewer to Section 3.2 and Appendix A (starting at line 743), which provides a theoretical justification for our proposed immunization technique, GIFT. Additionally, we experimentally compare against joint loss in Fig. 7, and empirically demonstrate that bi-level optimization works significantly better. This is consistent with findings in earlier immunization work: “We observe that direct maximization ruins the immunized model, i.e., low image quality
> when being adapted for another concept.” [1].
> ---
> 2. Our method does not require an explicitly curated or exhaustive “safe” dataset. Instead, similar to prior-preservation in DreamBooth, we generate the safe set on the fly using the base Stable Diffusion model before immunization. For example, when immunizing against “Van Gogh art,” we first sample images from related but non-malicious prompts such as “generic painting,” “art,” or “oil painting.” These samples provide sufficient coverage for preserving general visual capabilities without requiring an infinite or manually defined safe concept space.
> ---
> 3. We believe there may be a misunderstanding of our use case. GIFT is intended to be applied by model developers before releasing an open-weight diffusion model, not by end-users. Once immunized, the model behaves like the original model from a user perspective: it accepts text prompts normally and does not require fine-tuning for standard use. The only difference is that certain malicious concepts are made hard to re-learn through subsequent fine-tuning.
>
> - Our fine-tuning experiments are not meant to imply that users must fine-tune the model for normal operation. Instead, it is used solely to simulate attacker fine-tuning, as this is the threat model we evaluate, or normal fine-tuning on safe concepts. End-users can prompt the immunized model exactly as they would any Stable Diffusion checkpoint, or fine tune it, which is a common practice among the open source community, we just target making harmful fine-tuning hard/impractical. We demonstrate normal, prompt-based usability after immunization in Appendix D (Fig. 8).
>
> ---
> 4. In Figure 7, we show the qualitative results on the incremental addition of components in GIFT,  as stated in the caption. Column 3 corresponds to Maximization plus Prior, column 4 corresponds to Maximization plus Prior plus Representation Noising, and column 5 corresponds to the application of our bi-level optimization setup to the Maximization, Prior, and Representation Noising losses (i.e., full GIFT). There is no Maximization + Rep. Noise setting. In row 2 (malicious concept), GIFT produces the most degraded generations, which means it achieves the strongest suppression of the malicious concept, while still preserving safe concept generation in row 1. No ablated variant outperforms GIFT on the malicious concept. It is also worth noting that naively adding the prior loss on its own does in fact weaken the immunization as seen in Fig 7. The prior only works in the bi-level GIFT setting.
>
> - Regarding the visual difference between Figures 1 and 7, these figures use different concepts and prompts, and they probe different parts of the data distribution. Fine tuning or immunization are performed on top of a model that is itself trained on a particular data distribution, so the apparent strength of suppression for any single example will vary with concept complexity and the base model prior. This does not indicate weaker immunization. Our quantitative results, reported across many concepts, show that GIFT consistently suppresses malicious concepts while retaining safe content.
>
> ---
> [1] IMMA: https://arxiv.org/abs/2311.18815

---

### Official Review · Reviewer_Qswa · 2025-10-26

**Soundness:** 3
**Presentation:** 2
**Contribution:** 2
**Rating:** 4
**Confidence:** 3

**Summary:**

In this paper, the authors propose a gradient-aware immunization technique (GIFT) to defend diffusion models against malicious fine-tuning while preserving their ability to generate safe content. According to this approach, the immunization is formulated as a bi-level optimization process: the upper-level objective degrades the model’s ability to represent malicious concepts using representation noising and maximization, while the lower-level objective preserves performance on safe data. Extensive experiments on diverse concepts demonstrate that GIFT shows resistance against the re-learning of malicious content while successfully maintaining generation quality and preserving fine-tuning capabilities on safe data.

**Strengths:**

Here are the paper's strengths:
- It is well-written and well-documented
- The review of the state-of-the-art covers most of the relevant papers in the field
- The proposed approach is scientifically sound
- Provides comprehensive experiments across multiple domains (objects, art styles, NSFW content) with both qualitative and quantitative evaluation
- Experimental validation is complemented with ablation studies, prompt robustness tests, and analysis of hyperparameter effects (e.g., $\beta$)

**Weaknesses:**

Here are the paper's weaknesses:
- The proposed approach relies heavily on existing approach IMMA (Zheng & Yeh, 2024).
- Some aspects of the paper should be more clearly explained in order to improve readability, especially in section 3: the formalism is not sufficiently detailed
- Presentation is dense and sometimes awkwardly formulated (“of said parameters”  - lines 150-151, “LLM immunization technique” - line 200)
- The connection between theoretical motivation (MI reduction) and its practical implementation (representation noising loss) should be better explained
- The experimental validation is limited
- Some complexity analysis of the approach is missing, e.g. how the inner and outer loop steps affect training time or memory usage compared to fine-tuning

**Questions:**

Besides the weaknesses mentioned above, here is the rest of my concerns:
- The papers seem to rely heavily on IMMA (Zhang & Yeh, 2024). Therefore, the authors must critically discuss how their approach relates to IMMA's and clearly articulate their scientific contributions (apparently, IMMA also formulates immunization as a bi-level process).
- The formalism in section 3 should be significantly improved in order to increase readability. Sometimes the equations are just stamped in the paper without any justification.
- Some parameters are not explained. Who is $\psi'$ in eq. 2?
- How sensitive is your approach to the choice of $\alpha_I$ and $\alpha_P$ parameters. No discussion is provided in the paper regarding the choice of these parameters.
- Clarify how the alternating updates between safe and malicious datasets are scheduled during training?
- Why do you compare your approach only against IMMA and ESD? Please extend your comparison with other methods

---

> ### Author Response · Authors · 2025-11-21
>
> 1. We thank the reviewer for their comment, but we must respectfully disagree with the notion that our approach is reliant on IMMA. Although both GIFT and IMMA use a meta-learning setup, they differ fundamentally in objectives and behavior.
> - **Different intuition and optimization targets:** IMMA’s setup prevents malicious fine-tuning but causes substantial utility degradation (see Fig. 2, Fig. 6A, and Fig. 8). GIFT instead reformulates the meta-objective to explicitly retain utility on safe concepts through a second-order correction term (see Sec 3.2 and Appx A), enabling immunization that is aware of prior-preserving updates. This is an entirely different bi-level objective.
>
> - **Representation noising loss:** GIFT injects noise into intermediate representations of harmful concepts, making them harder to recover during future fine-tuning. This eliminates the need for concept-removal pre-processing steps (like ESD) required by IMMA (see Fig.7).
>
> - **Adaptor-agnostic:** IMMA must explicitly simulate a specific malicious adaptation method (e.g., DreamBooth, LoRA) and tailor immunization accordingly. GIFT learns to immunize without mimicking the attack, making it robust across different adaptation techniques without needing separate immunization runs.
>
> - **Better tradeoff:** GIFT consistently preserves safe-concepts to a far greater extent, making the model usable post-immunization. This distinction is critical for real-world deployment.
>
> - **Balancing retention and immunization is nontrivial:** Naive combinations of losses degrade either immunization or utility. GIFT achieves a non-obvious synergy in satisfying both objectives, which is a key technical contribution distinguishing it from prior work.
> ---
> 2. We thank the reviewer for recognizing that the paper is well-written and well-documented. We include a detailed explanation and mathematical derivations of our approach in Appx A, which extends our formulation in section 3. If there are specific equations that have not been sufficiently justified, we are happy to include further explanation in the paper.
> ---
> 3. We will also gladly use simpler phrases in presentation-dense areas. These are stylistic issues that we will revise for improved readability.
> ---
> 4. In section 3.3, we explained the move from the theoretical justification introduced by Rosati et al. (NeurIPS 2024) [1] to our current loss objective, but we can make this more explicit in revision if needed. By the data-processing inequality, the mutual information between the malicious input and the intermediate representation MI(xm; zm) is an upper bound on MI(xm; ym). See Eq. 6 and L197. This forms the basis of our approach: if we reduce the mutual information between the malicious input and the internal representations, then the mutual information between the malicious input and the output must decrease as well, making the harmful concept more difficult to recover during subsequent fine-tuning.
>
>     Directly optimizing MI is intractable for diffusion models, so we follow the insight of Rosati et al. (NeurIPS 2024) [1], who show that perturbing internal representations serves as an effective surrogate for MI reduction in practice. Our representation-noising loss implements this approach by injecting noise into activations associated with the malicious concept, explicitly reducing the information the model can rely on during later fine-tuning. Our ablations (Appx. Fig. 7) confirm that this loss is important for preventing malicious fine-tuning.
> ---
> 5. We thank the reviewer for noting that the experimental section is comprehensive (multi-domain, quantitative + qualitative, ablations, robustness, hyperparameter analysis). We believe this directly addresses the concern about limited experiments.
> ---
> 6. Running 1,500 steps of immunization with GIFT took 8.3 minutes and 26 GB of memory, compared to 7.3 minutes and 20.7 GB for standard fine-tuning. Both were run on an NVIDIA L4 (48 GB). GIFT adds only modest time (~12%) and memory overhead (~5GB).
> ---
> [1] Rosati et al: https://arxiv.org/abs/2405.14577

---

> > ### Author Response · Authors · 2025-11-21
> >
> > Regarding the rest of the questions not present in the weakness section:
> >
> > 1. $\psi$, in general, refers to the subset of model weights updated during the immunization process (see L136). Weights with a single prime mark (') denote parameters that have been updated by the prior-preservation objective; see the definition of $\theta'$ in Eq. (2). So, $\psi'$ is and updated $\psi$ after immunization. We are happy to clarify this more explicitly in the paper.
> > ---
> > 2. We selected the learning rates' starting points based on common practices for similar architectures [2]. We experimented with different rates, and we reported the ones that worked well for our experiments. That said, we agree that a comprehensive sensitivity study is a good addition.
> > ---
> > 3. In line 149 “To perform optimization, we compute parameters $θ^∗$ via a gradient step using the lower-level task on $D_S$ , followed by an optimization step of said parameters using the upper-level task on $D_M$.” Regarding alternating updates, it is a simple alternation between the 2 datasets.
> > ---
> > 4. We would like to point out that GIFT is specifically an immunization technique (preventing concepts from being re-introduced), rather than a concept erasure technique. At the time of submission, the only other immunization technique developed for diffusion models was IMMA. We only include ESD as a point of comparison between immunized and concept-erased models. It is known that other popular concept erasure methods (e.g., UCE, RECE) suffer from the same vulnerabilities as ESD [3]. We therefore feel that IMMA is the only technique that genuinely warrants a comparison to ours.
> > ---
> > [2] HuggingFace: https://huggingface.co/blog/dreambooth
> > [3] Pham et al: https://arxiv.org/abs/2308.01508

---

> > > ### Comment · Reviewer_Qswa · 2025-11-27
> > > **Official Comment by Reviewer Qswa**
> > >
> > > After reading the authors' rebuttal, I could say that my concerns have been partially addressed. The approach presented in the paper is largely incremental, and does not meet the typical level of novelty and rigor expected for a venue like ICLR. Therefore, I maintain my initial rating which is 4.

---

> > > > ### Author Response · Authors · 2025-11-28
> > > >
> > > > Thank you for the follow-up. We note that the reviewer states that their concerns were only partially addressed, but no additional questions or unresolved points are specified.
> > > >
> > > > In our rebuttal, we clarified point-by-point in detail how GIFT differs fundamentally from IMMA in objective, mechanism, theoretical grounding, and practical behavior, and explained why the contribution cannot be reduced to an incremental variant. We also addressed every technical question raised: formalism, notation, complexity, parameter choices, dataset scheduling, and conceptual clarity.
> > > >
> > > > While we respectfully acknowledge the reviewer’s final judgment, we believe that the rebuttal directly resolved all actionable concerns raised. If there are particular aspects that the reviewer feels remain unclear, we would be happy to further elaborate on them in the revision.

---

### Official Review · Reviewer_c2xJ · 2025-10-27

**Soundness:** 2
**Presentation:** 3
**Contribution:** 2
**Rating:** 2
**Confidence:** 4

**Summary:**

This paper introduces GIFT, a defense framework designed to protect text-to-image diffusion models from being maliciously fine-tuned to generate harmful or unauthorized content, while preserving their ability to generate safe content. GIFT formulates immunization as a bi-level optimization problem inspired by IMMA. The upper-level task maximizes a loss on malicious concepts combined with a “representation noising” term, while the lower-level task minimizes the prior-preservation loss on safe data.

**Strengths:**

1. The paper explores a valuable problem of making diffusion models resistant to malicious fine-tuning.
2. The authors structure the paper clearly.

**Weaknesses:**

The quality of the paper still needs improvement.
1. The experiments were only conducted on SD 1.5, lacking the latest diffusion models such as SD3, SD3.5, Flux, etc.
2. Regarding Figure 2, it appears that the CLIP Score of the SD model becomes lower than both GIFT and IMMA after fine-tuning for 1000 steps. Does this indicate that the evaluation prompts overlap too much with the training set, resulting in an effect where normal generation capability is maintained only on the evaluation prompts?
3. The evaluation metrics of the article include CLIP, etc., but lack safety-related evaluation metrics such as NudeNet and Q16.
4. Regarding robustness testing against malicious fine-tuning, there is a lack of security robustness tests, such as UnlearnDiff.
5. The dataset size is relatively small and lacks generalizability.
6. When evaluating normal images, relying solely on CLIP cannot fully reflect the generative capability for normal images. It is necessary to assess from various aspects such as aesthetics and image quality, rather than just the CLIP score.
7. There are too few diffusion model safe methods compared, with only ESD and IMMA being compared.

**Questions:**

See weaknesses.

**Details Of Ethics Concerns:**

No.

---

> ### Author Response · Authors · 2025-11-21
>
> 1. We agree that extending our method to newer architectures like SD3, SD3.5, and FLUX is important future work. However, we chose SD 1.5 as our main backbone for two reasons. First, it allows us to make a direct and fair comparison with existing immunization and concept erasure methods, such as IMMA and ESD. Second, recent empirical evidence shows that SD 1.x remains the dominant open-weight model family in real-world use. A recent study by Wagner et al. (2025) [2] reports that CivitAI (an extremely active platform for sharing and developing open-source T2I models) is still primarily populated by SD-based derivatives and adapters (mostly SD 1.5), with some SD 1.5 derivatives like Realistic Vision exceeding two million downloads (Figures 6 and 7). This suggests that SD 1.5 remains the backbone behind a large portion of real-world misuse.
> In principle, GIFT can be applied to SD3/3.5, FLUX, or similar models without modification because it does not assume any architectural structure. We view an evaluation on these newer families as valuable future work, but complementary to our core contribution, which is to show that we can robustly immunize the currently most widely deployed and misused open-weight backbone.
> ---
> 2. Could the reviewer please clarify the type of overlap referred to here?
> ---
> 3. We include safety-related evaluations in Appendix E, where we report NudeNet results in Table 2. We believe this section directly addresses the safety metrics the reviewer is referring to.
> ---
> 4. We believe this point is outside the main scope of our work. Our focus is on the effect of malicious fine-tuning on recovering immunized concepts, rather than jailbreak or prompting-based attacks such as those targeted by UnlearnDiff. We also evaluate GIFT’s robustness to different prompt rephrasings in Appendix G and show that rephrasing does not meaningfully weaken the immunization effect. If time and resources allow, we can include an additional experiment in the appendix, but we view this as complementary rather than essential to our core contribution.
> ---
> 5. Our evaluation spans three diverse domains: 26 object concepts from the Custom Concept 101 dataset (across families like toys, animals, household items, and more), 10 artistic styles (e.g., Van Gogh, Picasso), and 40 randomly sampled pornographic images from the NSFW-T2I dataset. We believe this setup offers broad and representative coverage. That said, we are open to including additional specific concepts or categories that the reviewer may consider essential for a more complete evaluation.
> ---
> 6. In addition to CLIP, we report LPIPS and DINO in several figures (Fig. 3, Fig. 5, Fig. 11, Fig. 12, Fig. 14) to provide a broader evaluation beyond CLIP.
>    - **LPIPS** measures perceptual similarity in feature space and aligns more closely with human judgments than pixel metrics.
>    - **DINO** evaluates high-level semantic similarity.
>
>    We also provide qualitative examples in Fig. 2, Fig. 6, Fig. 8, and Fig. 9.
>    We agree that developing dedicated evaluation metrics for this task remains an open challenge.
> ---
> 7. At the time of submission, IMMA was the only published method specifically addressing immunization of diffusion models, so we use it as our primary point of comparison. ESD is included as a standard concept-erasure baseline rather than a competing immunization technique. We did not include additional concept-removal methods because prior work has shown that these approaches are generally vulnerable to re-learning the removed concept after only a few fine-tuning steps [1]. For this reason, we focused our evaluation on the most relevant immunization method (IMMA) alongside a representative erasure baseline (ESD).
>
> ---
>
> **References**
>
> [1] Pham, Minh; Marshall, Kelly O.; Cohen, Niv; Mittal, Govind; Hegde, Chinmay. *Circumventing Concept Erasure Methods for Text-to-Image Generative Models*. arXiv:2308.01508 (2023).
> [2] Wagner, Laura; Cetinić, Eva. *Perpetuating Misogyny with Generative AI: How Model Personalization Normalizes Gendered Harm*. arXiv:2505.04600 (2025).

---

> > ### Comment · Reviewer_c2xJ · 2025-11-28
> >
> > Thanks for your rebuttal. I still have some concerns.
> > 1) I don’t think conducting experiments only on SD 1.x is sufficient, especially considering that many recent safety-related diffusion model papers since 2024 evaluate multiple versions of Stable Diffusion, such as safree [1] , meta-unlearning [2], PromptGuard [3], and so on.
> >
> > 2) Regarding the second question, it is noticeable that the yellow dashed line (SD) at 1000 steps lies mostly below the IMMA and GIFT trend lines. In other words, the original model’s CLIP score shows a larger drop, which seems to indicate the presence of bias in the training set.
> >
> > [1] Yoon, et.al. SAFREE: Training-Free and Adaptive Guard for Safe Text-to-Image And Video Generation. 2024
> > [2] Gao, et.al. Meta-Unlearning on Diffusion Models: Preventing Relearning Unlearned Concepts. 2025
> > [3] Yuan, et.al. PromptGuard: Soft Prompt-Guided Unsafe Content Moderation for Text-to-Image Models, 2025

---

> > > ### Author Response · Authors · 2025-12-03
> > >
> > > We thank the reviewer for the follow-up.
> > >
> > > 1. Our work targets the malicious fine-tuning threat model for open-weight diffusion models. Among such models, SD-1.x remains one of the most downloaded, redistributed, modified, and misused backbones in practice, making it a highly relevant target for reducing real-world risk. Even if a method is evaluated on a single model family, providing the first practical immunization mechanism for the model most widely involved in misuse is, we believe, a meaningful contribution.
> > >    That being said, GIFT model-agnostic and does not assume an architecture, and can be applied to newer diffusion families.
> > >    We also note that the methods cited by the reviewer (SAFREE, Meta-Unlearning, PromptGuard) study different threat models and are not evaluated under malicious fine-tuning. In addition, Meta-Unlearning and PromptGuard do not report experiments on SD3 or FLUX.
> > >
> > > 2. The dataset was not hand-selected. We used the Custom Concept 101 benchmark and included all 26 concepts that had at least eight images, which was our only criterion to ensure disjoint immunization and attack splits. Comprehensive results across all 26 objects are shown in Figure 3 and Figure 11 (Appendix I).
> > >    Regarding Figure 2, the observation that the SD baseline’s CLIP score falls below GIFT and IMMA after 1000 steps is consistent with overfitting. SD can overfit to the training dataset more quickly than the immunized models. This is also reflected in LPIPS, which increases for SD, indicating growing similarity to the training images. Thus, this behavior is expected and does not indicate bias or prompt overlap.

---

### Official Review · Reviewer_neBj · 2025-10-29

**Soundness:** 2
**Presentation:** 4
**Contribution:** 2
**Rating:** 4
**Confidence:** 4

**Summary:**

This paper presents GIFT, a post-training method for immunizing diffusion models against downstream fine-tuning on malicious concepts. GIFT alternates between two objectives: (1) a standard diffusion loss on safe concept images to preserve generation quality on safe concepts, and (2) a maximized diffusion loss on malicious concept images for immunization. Additionally, to prevent easy re-learning of malicious concepts, GIFT optimizes intermediate layer activations on malicious images to resemble noise sampled from the same distribution, encouraging the model to erase these concepts across all layers rather than only at the output, which could be easily re-adapted. GIFT is evaluated on immunization against specific objects, artistic styles, and NSFW content.

**Strengths:**

-  GIFT operates in post-training and can be applied to pre-trained models, thus not harming the pre-training process.

- The paper is clearly written and easy to follow.

- Enforcing the model to erase malicious concepts from all layers is interesting and the proposed approach is intuitive.

**Weaknesses:**

- **Immunization effectiveness.** Both figures 4 and 6 show that GIFT can still generate the immunized concepts in many of its outputs. This raises a concern about a simple attack: sampling multiple generations until the malicious concept appears. To address this, could the authors provide a larger set of uncurated generations (e.g., 25 samples) from a few randomly selected immunized concepts to better characterize the actual failure rate?

- **Limited concept coverage.** The paper demonstrates immunization on a small number of hand-selected concepts. To assess whether the method generalizes broadly, could the authors evaluate GIFT on a larger set of randomly sampled concepts with a few generations from each? For each concept, please show both attempted generations of the immunized malicious content and preserved safe content (as in Fig. 6).


- **Longer adaptation periods.** Does extended fine-tuning time enable re-learning of the immunized concepts? Figure 4 shows a decreasing CLIP score trend toward the end of training, which could support this hypothesis, though it may also indicate overfitting to the provided images. It would be valuable to evaluate: (1) CLIP scores over longer adaptation times, and (2) the LPIPS between generated images and their closest training image over time (i.e., compute LPIPS to each input image and report the minimum), to distinguish between overfitting and genuine concept recovery.

- **SD Performance.** Figure 8 shows that safe concept generation is noticeably degraded, with baseline SD producing more detailed outputs than SD+GIFT. This raises concerns about the practical utility of the method for preserving model performance on safe content.

Minor remarks:

- **Selected NSFW concept.** Figure 6 uses explicit pornographic scenes as the malicious concept. While I understand the need to demonstrate NSFW content erasure, other NSFW concepts (e.g., smoking) might demonstrate the same capability while being more appropriate for a broader reading audience.

- **Quantitative results.** Can the authors report average CLIP scores and minimum LPIPS to the closest input image at the end of fine-tunin, averaged over a large set of adaptations? This would provide a more comprehensive quantitative assessment of immunization effectiveness beyond the trajectory plots shown in the paper.

**Questions:**

- Can GIFT handle immunization to multiple different concepts at once? E.g., immunization to all NSFW concepts? Does performance on safe concepts  further degrades when GIFT is applied with multiple immunization concepts?

---

> ### Author Response · Authors · 2025-11-20
> **Response to Reviewer neBj (part 1)**
>
> **Immunization effectiveness**. We appreciate the reviewer’s concern about potential vulnerabilities to simple sampling attacks. However, the examples in Figures 4 and 6 primarily illustrate overfitting, not genuine retention of the immunized concept. For instance, in Figure 4, several generations resemble Van Gogh’s Starry Night even though the prompt was “a painting of a cat in Van Gogh style.” These images are visually “Van Gogh-like” but fail the actual prompt (i.e., they contain no cat). In contrast, outputs from ESD display more consistent visual alignment, **showing GIFT’s strong immunization effect**.
>
> To investigate this systematically and follow the reviewer’s suggestion, **we sampled 25 images from the GIFT-immunized model after fine-tuning on Van Gogh style and conducted a small blind survey** with seven annotators unaffiliated with the project, and we familiarized them with Van Gogh art style. We additionally included five control images from an unmodified Stable Diffusion model generated with the same prompt to validate annotation reliability. Annotators were asked to classify each image as success (contains a cat and reflects Van Gogh style), failure (missing one or both conditions), or not sure.
>
> All five control images were voted as successful. Among the 25 images from the immunized model, only 6 (24%) were labeled as successes, while 19 (76%) were labeled as failures. This indicates that while GIFT is not perfect, the majority of outputs don’t retain the targeted malicious concept, **meaning an adversary would face substantial overhead (e.g., repeated sampling and filtering) to extract useful content.**
>
> Overall, these findings support our claim that GIFT significantly increases the difficulty and cost of recovering the immunized concept, even under naive sampling-based attacks.
>
> **Limited concept coverage**. Our evaluation spans three diverse domains: **26 object concepts** from the Custom Concept 101 dataset (across families like toys, animals, household items, and more), **10 artistic styles** (e.g., Van Gogh, Picasso, etc.), and **40 randomly sampled pornographic images** from the NSFW-T2I dataset.
> We believe this setup offers broad and representative coverage. That said, we are open to including additional specific concepts or categories that the reviewer may consider essential for a more complete evaluation.
>
> **Longer adaptation periods.** Because CLIP measures the semantic similarity between an image and text, if it's decreasing, it means the model is not actually learning, which does not support the hypothesis that extended fine-tuning enables re-learning. We ran immunization for 1500 steps against Van Gogh art style. Then, we finetuned for 10k steps and reported average CLIP and minimum LPIPS in the table below.  We observe that the average CLIP score changes only slightly after 1000 steps (28.73 at 1000 steps versus 28.16 at 10000 steps), which suggests that extended adaptation does not significantly improve alignment with the Van Gogh style prompt. On the other hand, the **min_lpips is strictly decreasing which indicates that generations become increasingly similar to specific training images**. This pattern is consistent with overfitting to the provided images rather than genuine recovery of the immunized concept, even for much longer runs.
> | Steps | avg_clip | min_lpips|
> |-------|----------------|-----------|
> | 20    | 19.73          | 0.662     |
> | 500   | 22.06          | 0.583     |
> | 1000  | 28.73          | 0.579     |
> | 1800  | 27.64          | 0.228     |
> | 6000  | 28.36          | 0.235     |
> | 7000  | 26.48          | 0.134     |
> | 10000 | 28.16          | 0.087     |
>
> **SD Performance.** We acknowledge the degradation in safe concept performance shown in Figure 8. However, compared to IMMA, we believe **GIFT represents a significant leap forward in preserving safe content**. It is worth noting that degradation of benign concepts is a common issue in both immunization and concept erasure methods from published literature. This is also highlighted in recent work [1], which points to concept entanglement as a key challenge because modifying one concept can unintentionally impact others. While not fully solved yet, GIFT moves the field forward by offering a better tradeoff between immunization strength and safe content preservation.
>
> [1] Amara et al. Erasing More Than Intended? How Concept Erasure Degrades the Generation of Non-Target Concepts. arXiv preprint arXiv:2501.09833 (2025).

---

> > ### Author Response · Authors · 2025-11-21
> > **Response to Reviewer neBj (part 2) - Minor remarks and question**
> >
> > **Remark 1**: We appreciate the concern. We chose explicit NSFW content to test GIFT under the most challenging conditions. We already cover sensitive areas with black boxes, and we will enlarge them and clarify the nature of the content in the red notice at the beginning to ensure appropriateness for all readers.
> >
> > **Remark 2**:
> > Following the reviewer’s suggestion, we report the average CLIP score and minimum LPIPS at the end of fine-tuning (1500 steps), averaged over multiple concepts (26 objects and 10 artists). These results are consistent with the trends shown in Figure 5 and Figure 3: GIFT lies between IMMA and SD, reflecting the expected balance between immunization strength and safe-concept utility retention.
> >
> > | Model             | avg_clip @1500 | min_lpips @1500 |
> > |-------------------|----------------|------------------|
> > | **GIFT**          | 26.98          | 0.3558           |
> > | **IMMA**          | 26.40          | 0.4038           |
> > | **SD** | 27.52      | 0.1997           |
> > *Table 1: objects (26 concepts)*
> >
> > | Model             | avg_clip @1500 | min_lpips @1500 |
> > |-------------------|----------------|------------------|
> > | **GIFT**          | 30.92          | 0.4859           |
> > | **IMMA**          | 26.97          | 0.6003           |
> > | **ESD**           | 33.20          | 0.4567           |
> > *Table 2: Art styles (10 concepts)*
> >
> > **[Q] Can GIFT handle immunization to multiple different concepts at once?** Although GIFT was originally designed for single-concept immunization, we tested its ability to
> > handle multiple objects at once. Our results show that the immunized model successfully learned the safe concept, while it failed to learn the held-out malicious concept, as illustrated in Fig. 9. We include more details about this experiment in Appendix F.

---

### Note · Authors · 2026-02-01

I have read and agree with the venue's withdrawal policy on behalf of myself and my co-authors.

---

### Meta-Review · Area_Chair_KWXA · 2026-01-10

**Summary:**

While the authors have made an effort to address a portion of the concerns raised during the review process, there remain several critical issues that have not been sufficiently resolved. Regarding the effectiveness of the approach, I share the concerns of Reviewer neBj that the proposed method remains susceptible to attacks. Although the authors attribute successful attacks to overfitting, it is important to recognize that in the context of safety and robustness, testing against overfitting is often a necessary and encouraged part of a rigorous examination. Unfortunately, the experiments presented in the rebuttal appear to verify this vulnerability rather than alleviate the reviewers' apprehension.

Furthermore, on the aspect of definitions and scope, both Reviewer neBj and Reviewer HJV3 have pointed out that the paper demonstrates immunization only on a small set of hand-selected concepts, which limits the evaluation of the method's broader utility. This issue extends to the definition of "unsafe" content. While the authors argue in their response that they do not require an explicitly curated or exhaustive "safe" dataset, the definition of "safe" is intrinsically associated with the definition of "unsafe." Consequently, without a clear boundary, the scope of "unsafe" content remains effectively infinite, making the safety guarantees difficult to validate.

From a practicability standpoint, as noted by Reviewer neBj, the proposed method degrades the performance of normal, safe content generation. This is a significant drawback, as maintaining the quality of benign generation is crucial for the method's adoption.

Additionally, regarding the generic applicability of the method raised by Reviewer c2x, conducting experiments solely on Stable Diffusion 1.x is insufficient for a contemporary study. Considering the standards of recent safety-related diffusion model papers in 2024, an evaluation across multiple versions of Stable Diffusion is necessary to demonstrate generalization.

Finally, I concur with Reviewer Qswa that the readability of the paper requires improvement to clearly communicate the technical contributions.

Reviewer Qswa also rightly advised that more comprehensive parameter analysis is needed.

Beyond these specific points, there is an overarching concern that the technical connection between this paper and IMMA is excessively high.

Based on these cumulative factors, the paper does not yet meet the bar for acceptance.

**Reviewer Concerns:**

While the authors have made an effort to address a portion of the concerns raised during the review process, there remain several critical issues that have not been sufficiently resolved.

**Reviewer Scores:**

My impression is that most reviewers may maintain their current ratings, with the possibility of small adjustments. This assessment is made under the assumption that the evaluation process remains fair and focused on technical considerations.

---

### Decision · Program_Chairs · 2026-01-26

Reject